# PIVOTING THROUGH THE CHIRAL-CLOCK FAMILY

NICK G. JONES

*St John's College and Mathematical Institute, University of Oxford, UK*

ABHISHODH PRAKASH

*Rudolf Peierls Centre for Theoretical Physics, University of Oxford, UK*

PAUL FENDLEY

*All Souls College and Rudolf Peierls Centre for Theoretical Physics, University of Oxford, UK*

ABSTRACT. The Onsager algebra, invented to solve the two-dimensional Ising model, can be used to construct conserved charges for a family of integrable $N$-state chiral clock models. We show how it naturally gives rise to a "pivot" procedure for this family of chiral Hamiltonians. These Hamiltonians have an anti-unitary CPT symmetry that when combined with the usual $\mathbb{Z}_N$ clock symmetry gives a non-abelian dihedral symmetry group $D_{2N}$. We show that this symmetry gives rise to symmetry-protected topological (SPT) order in this family for all even $N$, and representation-SPT (RSPT) physics for all odd $N$. The simplest such example is a next-nearest-neighbour chain generalising the spin-$1/2$ cluster model, an SPT phase of matter. We derive a matrix-product state representation of its fixed-point ground state along with the ensuing entanglement spectrum and symmetry fractionalisation. We analyse a rich phase diagram combining this model with the Onsager-integrable chiral Potts chain, and find trivial, symmetry-breaking and (R)SPT orders, as well as extended gapless regions. For odd $N$, the phase transitions are "unnecessarily" critical from the SPT point of view.

## CONTENTS

## 1. INTRODUCTION

The interplay of symmetries, topology and entanglement results in a zoo of interesting topological phases of quantum matter [1, 2]. A particularly useful technique is to map a lattice model with well-understood physics to one with a non-trivial order. Kramers-Wannier (KW) duality [3], for example, relates models belonging to the trivial phase to those with spontaneously broken symmetries. In the setting of symmetry-protected topological (SPT) phases of matter, an analogous role is played by the so-called SPT entangler [4–8]. It is a finite-depth unitary operator that transforms trivial models into non-trivial SPTs characterised by unbroken symmetries and distinguished from the trivial phase by robust boundary modes and topological response to gauge fields.

*E-mail addresses*: `nick.jones@maths.ox.ac.uk`, `abhishodh.prakash@physics.ox.ac.uk`, `paul.fendley@physics.ox.ac.uk`.

The *pivot procedure* [7, 9] provides a systematic way of constructing such SPT entanglers. Pivot Hamiltonians generate SPT entanglers upon exponentiation, and themselves have long-range order. Adding them to trivial and SPT Hamiltonians produces a rich phase diagram. Analysing this structure allows a deeper understanding of how various quantum orders are intertwined in the phase diagrams of lattice models. The key example of [7, 9] starts with transverse-field Ising chain (i.e. a qubit chain) and yields the cluster model Hamiltonian [10] describing a non-trivial SPT phase [11].

We show how the pivoting in this case follows directly from the *Onsager algebra*, introduced by Onsager to compute the free energy of the 2D classical Ising model [12] and the spectrum of the corresponding quantum chain. This infinite-dimensional Lie algebra is constructed by splitting the transverse-field Ising Hamiltonian into two pieces $A_0$ and $A_1$, the former coupling to the transverse field and the latter the nearest-neighbour interaction. As we explain below, the other generators are found by taking repeated commutators with these two [13, 14], and imposing the Dolan-Grady relations [15].

The Onsager algebra is all that is needed to implement the pivot procedure. Thus any Hamiltonians giving rise to this algebra satisfy the same pivot relations. A beautiful such set of spin chain models occurs in $\mathbb{Z}_N$-invariant clock chains [13, 16, 17]. Namely, the "superintegrable chiral Potts" chain [18] also can be split into two pieces that generate the identical algebra. For $N > 2$ the algebra is not sufficient to solve the model, but it can be used to construct a sequence of commuting charges. These charges indicate the chain with periodic boundary conditions is integrable.

In this paper, we exploit this connection to apply the pivot procedure to this family of Onsager-integrable clock Hamiltonians. The resulting Hamiltonians are related by both pivoting and Kramers-Wannier duality, as illustrated in Fig. 1. We show that the global symmetry for a given $N$ is the dihedral group $D_{2N}$, arising from the $\mathbb{Z}_N$ clock symmetry along with an anti-unitary CPT symmetry. One key consequence of the pivot procedure is that non-trivial models (for example with symmetry

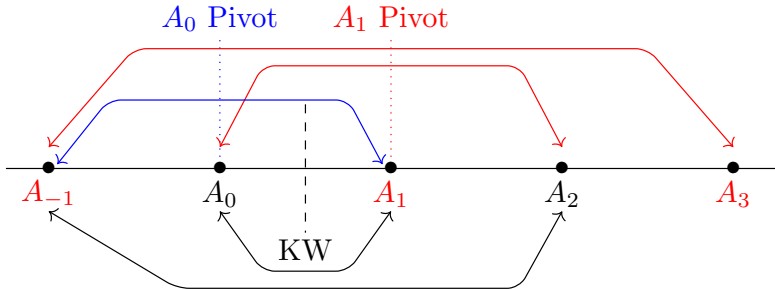

FIGURE 1. Web of maps generated by pivoting and Kramers-Wannier duality in the Onsager-integrable chiral-clock family.

fractionalisation) can arise by unitarily transforming from models with easy-to-understand behaviour. For example, taking $A_0$ to be the Hamiltonian yields a trivial paramagnet, while taking $A_1$ induces spontaneous symmetry breaking. The Hamiltonian $A_2$ is generated by pivoting $A_0$ with $A_1$. For $N = 2$ these correspond to the familiar Ising paramagnet and ferromagnet respectively. while $A_2$ is the cluster Hamiltonian with SPT order (see Eq. (6) below).

One of the main results of this paper is a demonstration that for any even $N$, $A_2$ describes an SPT phase. For odd $N$, it describes a "representation SPT" (RSPT) [19], similar to what occurs in even-spin Haldane phases [20]. RSPTs do not enjoy the same topological protection as SPTs, but nevertheless have some similar phenomenology [21].

Both SPT and RSPT phases exhibit symmetry fractionalisation, but only for the former are the representations projective. The dominant Schmidt eigenvalues for both form a doublet, but only for the former do all eigenvalues pair. Thus the SPT at even $N$ disappears only when parameters are tuned through a bulk phase transition. For odd $N$, however, the RSPT can disappear [21] at a boundary transition or at an "unnecessary" bulk transition [22–28]. Connecting the various Hamiltonians yields interesting phase diagrams. We find (R)SPT phases and extended gapless phases as well as more

conventional disordered and spontaneous symmetry broken phases. The interpolation $H = A_1 + \lambda A_0$ is the familiar transverse-field Ising model for $N = 2$ and corresponds to the "superintegrable chiral Potts" chain for $N \geq 3$ [16,29]. The latter, at least for $N = 3$, contains extended gapless phases [30,31]. Hamiltonians that connect $A_0, A_1, A_2$ exhibit an even richer phase diagram, for example including symmetry-enriched critical points for $N = 2$ [6,32]. Another result of ours is to show that for $N = 3$ and $N = 4$ this model hosts a variety of strongly correlated gapped and gapless states. For any linear combination of the $A_k$, the model is presumably integrable, as it possesses an infinite set of local and mutually-commuting charges [17]. However, utilising integrability in chiral clock models requires a rather intricate analysis, and we defer it to a separate paper [33].

The paper is organised as follows. In Section 2, we introduce both the pivot procedure and the Onsager algebra, and show that implementing the former follows directly from building Hamiltonians from the generators of the latter. In Section 3 we introduce the chiral-clock family of Hamiltonians and its Onsager-algebra structure. We find its symmetries and ground states, in particular the matrix-product state ground state for $A_2$. In Section 4 we show that SPT and RSPT phases occur at even and odd $N$ respectively, all exhibiting symmetry fractionalisation. In Section 5 we discuss the phase diagram of the combined Hamiltonian $\alpha A_0 + \beta A_1 + \gamma A_2$, illuminating how the distinct phases fit together. Finally, we detail some natural questions for future work.

## 2. Pivoting with Onsager

2.1. **What Onsager did.** Onsager's solution of the two-dimensional classical Ising model with periodic boundary conditions [12] is a tour de force. He computed the full spectrum of the transfer matrix, and hence the exact partition function. Taking a strongly anisotropic limit then yields the exact spectrum of the corresponding quantum Hamiltonian of the transverse-field Ising chain.

The core of Onsager's result comes from understanding the algebra obeyed by the generators of the Hamiltonian and transfer matrix. These generators are written in terms of Pauli matrices acting on the usual Hilbert space of $L$ two-state systems, i.e. $(\mathbb{C}^2)^{\otimes L}$. The two basic generators are

$$A_0 = -\sum_{j=1}^{L} \sigma_j^x \ , \qquad A_1 = -\sum_{j=1}^{L} \sigma_j^z \sigma_{j+1}^z \ , \tag{1}$$

where in the latter $\sigma_{L+1}^z \equiv \sigma_1^z$. The transverse-field Ising Hamiltonian with periodic boundary conditions is then simply $H_{\mathrm{I}} = A_0 + \lambda A_1$.

Other generators of the Onsager algebra are found by taking commutators of the basic two, subject to two key identities. Onsager showed by explicit computation that

$$\Big[A_1, \big[A_1, [A_1, A_0]\big]\Big] = 16[A_1, A_0], \qquad \Big[A_0, \big[A_0, [A_0, A_1]\big]\Big] = 16[A_0, A_1]. \tag{2}$$

Imposing these identities, now known as the Dolan-Grady conditions, results in an infinite-dimensional Lie algebra with generators $\{A_l, G_m | l, m \in \mathbb{Z}\}$, with $G_{-k} = -G_k$ and $G_0 = 0$. The next two generators are defined by

$$G_1 = \tfrac{1}{2}\big[A_1, A_0\big] \ , \qquad A_2 = A_0 - \tfrac{1}{4}\big[A_1, G_1\big] \tag{3}$$

so that the first Dolan-Grady condition can be written as $\big[A_1, A_2\big] = \big[A_0, A_1\big]$. With $A_0$ and $A_1$ defined by (1), $A_2 = \sum_j \sigma_{j-1}^z \sigma_j^x \sigma_{j+1}^z$ results. (Here and henceforth all sums over $j$ run from 1 to $L$.) Proceeding in this fashion, Onsager defined and found explicit expressions for all the generators, and showed that they satisfy

$$\boxed{\big[A_l, A_m\big] = 4G_{l-m} \ , \qquad \big[G_l, A_m\big] = 2A_{l+m} - 2A_{m-l} \ , \qquad \big[G_l, G_m\big] = 0 \ .} \tag{4}$$

A quicker way of establishing (4) for $A_0$ and $A_1$ from (1) is by rewriting them in terms of Majorana-fermion bilinears by using the Jordan-Wigner transformation. A commutator of fermion bilinears always yields a bilinear, so all the Onsager generators in this representation can be written as fermion bilinears. Explicit expressions and other useful details may be found in e.g. [34].

The exact spectrum of $H_{\mathrm{I}}$ follows from the Onsager algebra (4) because of a crucial simplification of the presentation (1). As there are only $L(2L-1)$ possible fermion bilinears on $L$ sites, there can be at most $2L-1$ distinct generators of (4) (by construction all generators are translation invariant). Indeed, $A_{l+L} = \pm A_l$ and $G_{l+L} = \pm G_l$, where the sign can be found in [12] (or working out the explicit fermion-bilinear expressions). To find the spectrum, one then can reduce the spectrum of $H_{\mathrm{I}}$ to a sum by taking the Fourier transformation of the Onsager generators.

Such a simplification does *not* occur in the chiral-clock presentations of the Onsager algebra that we study. As we describe below, however, the Onsager algebra provides a systematic way of constructing pivot Hamiltonians, making it useful in any presentation.

### 2.2. The Ising pivot.

The pivot procedure provides a method for generating SPT phases via unitary transformations [7,9]. One starts with a Hamiltonian $H_0$ with a trivial ground state and then searches for a local "pivot" Hamiltonian $H_{\mathrm{pivot}}$ that yields an SPT Hamiltonian via

$$H_{\mathrm{SPT}} = U(\pi)H_0 U(\pi)^\dagger \,, \qquad U(\theta) = \exp(-i\theta H_{\mathrm{pivot}}) \,. \tag{5}$$

The simplest example comes from the Ising Hamiltonian (1). Taking $H_0 = A_0$ from there gives a trivial paramagnetic phase with a unique ground state. Defining $H_{\mathrm{pivot}} = A_1/4$ from (1) then gives

$$H_{\mathrm{SPT}} = e^{-i\pi A_1/4} A_0\, e^{i\pi A_1/4} = \sum_j \sigma^z_{j-1}\sigma^x_j\sigma^z_{j+1} = A_2 \,. \tag{6}$$

All three of these Hamiltonians have a $\mathbb{Z}_2 \times \mathbb{Z}_2^T$ symmetry, generated by the spin-flip $\prod_j \sigma^x_j$, and complex conjugation in the $Z$-diagonal basis. This symmetry protects the SPT order.

The Hamiltonian $-H_{\mathrm{SPT}}$ is the canonical cluster model [10], and both $\pm H_{\mathrm{SPT}}$ exhibit SPT order. (The sign can be toggled by conjugating by $\prod_k \sigma^x_{4k}\sigma^x_{4k+1}$ [7].) Stated differently, $\exp(-i\pi A_1/4)$ is the cluster SPT entangler[1]. Since the ground state of $A_0$ is a product state and the terms in $A_1$ are all mutually commuting, the ground-state of $A_2$ can be written as a matrix-product state (MPS) with bond dimension two. Continuing in this fashion allows us to generate an infinite family called the (generalised) cluster models [5,40–45]. For example, pivoting $H_{\mathrm{SPT}}$ with $H_0$ gives another (SPT) spin chain with Hamiltonian $\sum_j \sigma^y_{j-1}\sigma^x_j\sigma^y_{j+1}$. Since $A_0$ and $A_1$ from (1) can be written in terms of fermion bilinears, all Hamiltonians generated in this fashion are written as bilinears. They all can thus be easily solved via standard techniques.

In general, the pivot procedure [7] works as follows. The starting point is a Hamiltonian $H_0$ with symmetry group $G$, and a product state for its unique ground state. Considering the one-parameter deformation $H(\theta) = e^{-i\theta H_{\mathrm{pivot}}} H_0 e^{i\theta H_{\mathrm{pivot}}}$, we *require* that $H(2\pi) = H(0)$, and that $H(\pi)$ is a non-trivial SPT phase protected by the group $G$. This means that, for $U(\theta) = \exp(-i\theta H_{\mathrm{pivot}})$, $U(2\pi)$ is a symmetry of $H_0$, while $U(\pi)$ acts as an SPT entangler.

### 2.3. Pivoting with Onsager.

Here we show that constructing pivots of this type does *not* require the explicit representation (1), but only the Onsager algebra. Given *any* lattice Hamiltonian realisation of $A_0$ and $A_1$ satisfying the Dolan-Grady relations, we can generate a family of Hamiltonians that behave nicely under pivoting. We here prove this fact, and in the rest of the paper exploit it.

The connection between the two procedures is rather direct. Indeed, when $A_0$ and $A_1$ are defined by (1), $A_2$ from the Onsager definition (3) is equal to $H_{\mathrm{SPT}}$ from (6). The latter equation thus implies that $A_2$ can be found by pivoting with $A_1/4$, i.e.

$$A_2 = e^{-i\pi A_1/4} A_0\, e^{i\pi A_1/4} \,. \tag{7}$$

This relation follows directly from the Onsager algebra, and is a particular case of the general identity (see [13,46] for related results)

$$e^{-i\alpha A_m} A_l\, e^{i\alpha A_m} = \cos^2(2\alpha)\, A_l + \sin^2(2\alpha)\, A_{2m-l} + i\sin(4\alpha)G_{l-m}. \tag{8}$$

---

[1]The fact that the ground state of the cluster model can be generated via a finite-time evolution generated by Ising-like Hamiltonians is well known from the study of measurement-based quantum computing (MBQC) [35]. In this context the cluster state, as well as other SPT ground states, serves as a good resource state [36–39].

To prove (8), we first use the Onsager algebra to generalise the Dolan-Grady relations to

$$\Big[\big[[A_l, A_m], A_m\big], A_m\Big] = 4\Big[[G_{l-m}, A_m], A_m\Big] = 8\big[A_l - A_{2m-l}, A_m\big] = 32\, G_{l-m} = 16\big[A_l, A_m\big] \ . \qquad (9)$$

Using the standard identity [47]

$$e^{-B} C e^{B} = \sum_{p=0}^{\infty} \frac{1}{p!} \underbrace{\Big[[C, B], B], \dots B\Big]}_{p-\text{fold}} \qquad (10)$$

with (9) yields

$$e^{-i\alpha A_m} A_l e^{i\alpha A_m} = A_l + \big[A_l, A_m\big] \sum_{n=1}^{\infty} \frac{(i\alpha)^{2n-1}}{(2n-1)!} 16^{n-1} + \big[[A_l, A_m], A_m\big] \sum_{n=1}^{\infty} \frac{(i\alpha)^{2n}}{(2n)!} 16^{n-1}.$$

The two sums are $i \sin(4\alpha)/4$ and $-\sin^2(2\alpha)/8$ respectively, and using (4) to rewrite the remaining commutators yields (8).

The identity (8) yields a sequence of exact pivot relations, with no further computations necessary. Defining $U_m(\theta) = e^{-i\theta A_m/4}$ yields

$$U_m(\pi) A_l\, U_m(\pi)^{\dagger} = A_{2m-l} \ , \qquad U_m(2\pi) A_l\, U_m(2\pi)^{\dagger} = A_l. \qquad (11)$$

All $A_l$ thus can be generated by a sequence of pivots with $A_0$ and $A_1$:

$$A_{l+2} = U_1(\pi) U_0(\pi) A_l\, U_0(\pi)^{\dagger} U_1(\pi)^{\dagger} \ , \qquad A_{-l} = U_0(\pi) A_l\, U_0(\pi)^{\dagger} \ . \qquad (12)$$

The $A_l$ thus fall into an even and an odd family, unitarily equivalent to $A_0$ and $A_1$ respectively. The two families are themselves related by KW duality (see Fig. 1). Moreover, any unitary symmetry commuting with $A_0$ and $A_1$ commutes with all the $A_l$. The same holds for anti-unitary symmetries, because using $U_m(-\pi)$ in Eq. (11) yields the same action on the $A_l$.

This construction immediately gives us information about the phase structure. Suppose that $A_0$ is a representative of the trivial phase. Then $A_{2k}$ for all $k$ necessarily has a unique ground state, and if $A_{2k}$ is a non-trivial SPT for any value of $k$, Eq. (11) tells us that $A_k/4$ is a pivot Hamilton giving the SPT entangler for this model. Moreover, the pivot Hamiltonian itself generates a symmetry of the 'halfway point' between the starting model and the pivoted model:

$$\big[A_l,\, A_{l-m} + A_{l+m}\big] = 4G_{-m} + 4G_m = 0. \qquad (13)$$

In our examples, all $A_l$ have integer eigenvalues and so generate a $U(1)$ symmetry.

## 3. THE INTEGRABLE CHIRAL CLOCK CHAINS

In this section we give a set of chiral Hamiltonians satisfying the Onsager algebra, and explore their basic properties using pivoting. We exploit the fact that the Onsager algebra automatically follows from any $A_0$ and $A_1$ obeying Dolan-Grady conditions; the remainder of the relations are simply definitions and consistency conditions [13–16]. Thus any Hamiltonian built from Onsager-algebra generators $A_l$ possesses an elegant pivot structure, i.e. any model obeying (2) automatically satisfies (11). We emphasise that although the chains are integrable, they are *not* free-fermionic.

3.1. **Onsager and the integrable chiral clock models.** We study Hamiltonians generated by two pieces of the "superintegrable chiral Potts" Hamiltonian chain [13, 14, 16, 17]. These pieces satisfy the Dolan-Grady relations and hence generate the Onsager algebra. We call this set of chains the Onsager-integrable chiral-clock family.[2] The Hilbert space is a chain of $N$-state quantum systems, i.e. $\left(\mathbb{C}^N\right)^{\otimes L}$, acted on by "shift" and "clock" operators generalising Pauli matrices. Each such operator

---

[2]We call them "clock" instead of "Potts" chains because the latter typically have $S_N$ symmetry that ours do not possess. We use "Onsager-integrable" instead of "superintegrable" as the former is more specific.

$X_j$, $Z_j$ acts non-trivially on a single site $j$ of the chain, and they obey $X_j Z_k = \omega^{\delta_{jk}} Z_k X_j$, along with $(X_j)^N = (Z_j)^N = 1$. In the $Z$-diagonal basis they act on the $j$th site as

$$X_j = \sum_{a_j=0}^{N-1} |a_j - 1\rangle \langle a_j| \qquad\qquad Z_j = \sum_{a_j=0}^{N-1} \omega^{a_j} |a_j\rangle \langle a_j| \qquad (14)$$

while leaving other sites unchanged. We have defined $\omega = e^{2\pi i/N}$ and identify basis states modulo $N$, i.e. $|a\rangle \equiv |a \bmod N\rangle$. For $N = 2$ they reduce to the corresponding Pauli operators $\sigma_j^x$, $\sigma_j^z$.

We build our Hamiltonians from the operators

$$h_{2j} = X_j , \qquad h_{2j-1} = Z_{j-1}^{-1} Z_j \qquad (15)$$

where the site-index $j$ on the right-hand-side is, as always, defined mod $L$. These operators obey $\omega h_k h_{k+1} = h_{k+1} h_k$ and commute otherwise. The Onsager generators are then

$$A_0 = -\frac{4}{N} \sum_j \sum_{m=1}^{N-1} \alpha_m (h_{2j})^m , \qquad A_1 = -\frac{4}{N} \sum_j \sum_{m=1}^{N-1} \alpha_m (h_{2j-1})^m , \qquad \alpha_m = \frac{1}{1 - \omega^m}. \qquad (16)$$

For $N = 2$ these reduce to (1). This presentation is "self-dual" in that $A_0$ and $A_1$ are related by Kramers-Wannier duality. This duality here shifts all $h_k \to h_{k+1}$ and so exchanges $A_0$ and $A_1$. Since the algebra of the $h_m$ is invariant under this shift, one Dolan-Grady condition implies the other. The "superintegrable chiral Potts" Hamiltonian is $H(\lambda) = A_1 + \lambda A_0$; it is an anisotropic limit of the 2D classical "chiral Potts" model [13, 18, 48, 49].

The operators from (16) satisfy the Dolan-Grady conditions (2) and hence generate the full Onsager algebra (4) for any $N$ [16]. We review this calculation in Appendix A. Closed-form expressions for the generators in general, however, are not known, as the explicit expressions get rather nasty beyond the first few. Expressions for $A_{-1}$ can be found in [17, 34]. The expression for $A_2$ is found by using duality from $A_{-1}$, pivoting using (7), or simply working out the commutators from the definition (3). We present the calculations in Appendix A; see also [13]. The nicest way to write the result is as

$$A_2 = -\frac{4}{N} \sum_j \sum_{m=1}^{N-1} \alpha_m S_{j-1,j}^{(m)} X_j^m S_{j,j+1}^{(m)} ,$$

$$S_{j-1,j}^{(m)} = 1 - \frac{2m}{N} - \frac{2}{N} \sum_{m'=1}^{N-1} \alpha_{m'} (1 - \omega^{mm'}) Z_{j-1}^{-m'} Z_j^{m'} . \qquad (17)$$

A key feature of this form is that $S_{j-1,j}^{(m)}$ has eigenvalues $\pm 1$. Thus despite its complicated-looking definition, in the $Z$-basis this operator is diagonal with entries $\pm 1$. The expression for $A_{-1}$ is found simply by writing these expressions in terms of the $h_k$ and then shifting $h_k \to h_{k+1}$.

The Onsager relations mean that any linear combination of the $A_l$ possesses an infinite sequence of local and mutually commuting charges [17], i.e.

$$\mathcal{H} \equiv \sum_{k=a}^{b} t_k A_k , \qquad Q_m \equiv \sum_{k=a}^{b} t_k (A_{m+k} + A_{-m+k}) \quad \Longrightarrow \quad [\mathcal{H}, Q_m] = 0 . \qquad (18)$$

The existence of this sequence implies that any such model is integrable. These conserved charges do not exhaust the symmetries of $\mathcal{H}$. A $U(1)$-invariant Hamiltonian that commutes with all the Onsager generators was discussed in depth in [34]. Thus $\mathcal{H}$ commutes with this Hamiltonian, meaning the latter can be thought of as an additional conserved charge. Another symmetry is the dihedral symmetry discussed next.

3.2. **Dihedral symmetry.** The appearance of a larger non-abelian symmetry group in chiral clock models is known; see e.g. [31, 50, 51]. We describe its most general form here. The generators of the Hamiltonians are all invariant under the $\mathbb{Z}_N$ symmetry generated by

$$r = \prod_j X_j \qquad \Longrightarrow \qquad h_k r = r h_k . \qquad (19)$$

Less obviously, the symmetry extends to the dihedral group $D_{2N}$, whose generators obey

$$D_{2N} \cong \langle r, s | r^N = s^2 = 1, \ srs = r^{-1} \rangle. \tag{20}$$

The second generator $s$ implements CPT symmetry here. We define $P$ to be spatial inversion, exchanging site $j$ with $L + 1 - j$. (The particular choice of fixed site or bond is arbitrary in a translationally invariant system.) Charge conjugation $C$ is defined so that $CX_jC^\dagger = X_j^\dagger$ and $CZ_jC^\dagger = Z_j^\dagger$ for all $j$. In the $Z$-diagonal basis it is

$$C = \prod_j C_j, \qquad \text{where} \quad C_j = \sum_{a=0}^{N-1} |a_j\rangle \langle N - a_j|. \tag{21}$$

Time reversal is implemented by an anti-unitary operator $\mathcal{K}$ that we define as complex conjugation in the $Z$-basis. It is simple to check that $s = CP\mathcal{K}$ is a symmetry of all our chiral-clock Hamiltonians:

$$sA_0s = A_0, \qquad sA_1s = A_1 \qquad \Longrightarrow \ sA_ls = A_l. \tag{22}$$

The dihedral symmetry will prove crucial in our analysis of the phases of these Hamiltonians.

3.3. **Maps amongst the family.** One key observation in Ref. [7] is that the pivoting relation in the cluster models gives us a large number of mappings between models. Since we showed that the unitary transformations (11) follow solely from the Onsager algebra, pivoting with $A_m$ thus unitarily transforms any $A_l \to A_{2m-l}$. Thus visualising the space of models $\{A_k\}$ as points on a line, pivoting with $A_m$ corresponds to reflection around each point $m$. Combining two pivots as in e.g. (12) gives a unitary transformation that shifts the index $A_l \to A_{l+2}$, as illustrated in Fig. 1.

As indicated above, Kramers-Wannier duality maps $A_0 \to A_1$ and vice versa. The Onsager algebra then requires that sending $A_0 \to A_1$ maps $A_n \to A_{1-n}$. In Fig. 1, this map corresponds to reflection about the bond between $A_0$ and $A_1$. Some care must be taken: Kramers-Wannier duality is not invertible, and so is not a one-to-one map. Indeed, we show explicitly below that the ground state of $A_1$ (and hence all $A_l$ for odd $l$) is $N$-fold degenerate, while the ground state of $A_0$ is unique.

Other operators allow us to relate different models. The anti-unitary operator $\mathcal{V} = \prod_j Z_j\mathcal{K}$ obeys

$$\mathcal{V}^2 = (-1)^L, \qquad \mathcal{V}A_{2k+1} = A_{2k+1}\mathcal{V}, \qquad \mathcal{V}A_{2k} = -A_{2k}\mathcal{V}. \tag{23}$$

There is a unitary operator with the same commutation/anticommutation property [31]. When $L = 0 \bmod N$ the unitary operator $\mathcal{W} = P\prod_j X_j^j$ obeys [31]

$$\mathcal{W}^2 = 1, \qquad \mathcal{W}A_{2k+1} = -A_{2k+1}\mathcal{W}, \qquad \mathcal{W}A_{2k} = A_{2k}\mathcal{W}. \tag{24}$$

Combining the two shows that the spectrum of any linear combination of the $A_l$ is symmetric about zero when $L$ is a multiple of $N$. Moreover, the spectrum of $\mathcal{H} = \sum_l t_lA_l$ is invariant under sending all $t_{2k} \to -t_{2k}$.

3.4. **Ground states.** Thanks to the Onsager algebra and the pivot relations, determining the ground state(s) of any Hamiltonian $A_l$ is straightforward.

Although they look rather complicated in their definition, the operators $A_0$ and $A_1$ individually take on a simple form in the right basis [13, 16]. The eigenvectors of $X_j$ are

$$\left| v_j^{(n)} \right\rangle = \frac{1}{\sqrt{N}} \sum_{a_j=0}^{N-1} \omega^{-na_j} |a_j\rangle \quad \Longrightarrow \ X_j \left| v_j^{(n)} \right\rangle = \omega^{-n} \left| v_j^{(n)} \right\rangle, \tag{25}$$

so the eigenvectors of $A_0$ are simply product states

$$A_0 \prod_j \left| v_j^{(n_j)} \right\rangle = E_0(\{n_j\}) \prod_j \left| v_j^{(n_j)} \right\rangle, \qquad \text{where} \quad E_0(\{n_j\}) = -\frac{4}{N} \sum_{j=1}^{L} \sum_{m=1}^{N-1} \alpha_m\omega^{-mn_j} \tag{26}$$

for any choice of the $n_j = 0 \ldots N - 1$. The eigenvalues simplify using the trigonometric identity

$$\sum_{m=1}^{N-1} \alpha_m\omega^{-mn} = \frac{(N-1)}{2} - n, \qquad 0 \le n \le N - 1, \tag{27}$$

so that

$$E_0(\{n_j\}) = -2L\frac{N-1}{N} + \frac{4}{N}\sum_j n_j \ . \tag{28}$$

Hence the unique ground state of $A_0$ has all $n_j = 0$, yielding a trivial paramagnet. Worth noting is that the full spectrum is invariant under $E_0 \to -E_0$, and that all eigenvalues are integers up to the shift and the overall factor of $4/N$.

Any basis state in the $Z$-diagonal basis is an eigenstate of $A_1$. Denoting the eigenvalue of $Z_j$ on each site as $\omega^{a_j}$, using (27) gives the eigenvalue of $A_1$ to be

$$E_1 = -2L\frac{N-1}{N} + \frac{4}{N}\sum_j \Big((a_j - a_{j+1})\bmod N\Big). \tag{29}$$

We emphasise that each term in this sum is taken mod $N$, as a consequence of the restriction in (27). The energy $E_1$ from (29) is invariant under shifting all $a_j \to (a_j + m)\bmod N$ for any $m$, so each level is $N$-fold degenerate. The $N$ ground states of $E_1$ are therefore given by setting $a_j = a$ for all $j$. These ferromagnetic ground states spontaneously break the $\mathbb{Z}_N$ symmetry $r$.

The operators $A_l$ for even and odd $l$ are unitarily equivalent to $A_0$ and $A_1$ respectively, as follows from the pivot relation (12). Thus $A_l$ has a unique ground state for $l$ even, while for odd $l$ it has an $N$-fold ground-state degeneracy. Moreover, since $U_0$ is a product of on-site unitary operators, it can be thought of as a matrix-product unitary operator (MPU) of bond-dimension zero. For the operator $U_1$, we exploit the fact that $A_1$ is a sum of commuting terms. Then $U_1(\pi) = \exp(-i\pi A_1/4)$ can be written as a product of two-site unitaries as

$$U_1(\pi) = \prod_j U_{j,j+1}, \qquad U_{j,j+1} \equiv \exp\left(i\frac{\pi}{N}\sum_{m=1}^{N-1}\frac{1}{1-\omega^m}Z_j^{-m}Z_{j+1}^m\right). \tag{30}$$

As illustrated in Fig. 2, we can rewrite this product as an MPU of bond-dimension $N$. Thus the ground states of $A_{2k}$ and $A_{2k+1}$ can each be written as an MPS of bond dimension upper bounded by $N^k$.

FIGURE 2. Graphical representations of $U_1(\pi)$ as products $U_{j,j+1}$ from (30). The left-hand picture is a depth-two local unitary circuit, the middle a staircase circuit. The latter can be interpreted as an MPU with bond-dimension $N$ (right).

The $N$-channel MPS for the ground state of $A_2$ can be put in an elegant form. Using a bit of Fourier transformation along with (27) shows that the MPU tensor acts on the $X$-basis eigenstates (25) as

$$U_{j,j+1}\left|v_j^{(s)}v_{j+1}^{(t)}\right\rangle = \frac{1}{N}\sum_{r=0}^{N-1}\lambda_r\left|v_j^{(s+r)}v_{j+1}^{(t-r)}\right\rangle, \qquad \lambda_r = \frac{\omega^{r/2}}{\sin(\pi(r+\frac{1}{2})/N)} \ . \tag{31}$$

The ground state of $A_2$ is thus

$$\begin{aligned}
|\psi_2\rangle &= U_{L1}\cdots U_{23}U_{12}\left|v_1^{(0)}v_2^{(0)}\cdots v_L^{(0)}\right\rangle \\
&= N^{-L}\sum_{n_1,\dots,n_L}\left(\prod_{j=1}^L\lambda_{n_j}\right)\left|v_1^{(n_1-n_L)}v_2^{(n_2-n_1)}v_3^{(n_3-n_2)}\cdots v_L^{(n_L-n_{L-1})}\right\rangle \\
&= N^{-3L/2}\sum_{a_1,\dots,a_L}\sum_{n_1,\dots,n_L}\left(\prod_{j=1}^L\lambda_{n_j}\omega^{a_j(n_{j-1}-n_j)}\right)|a_1\cdots a_L\rangle
\end{aligned} \tag{32}$$

where all sums run from 0 to $N-1$. Converting the sums over the $\{n_j\}$ into matrix products yields

$$|\psi_2\rangle = \sum_{a_1,\ldots,a_L} \text{tr}\big(\mathcal{A}_{a_1}\cdots\mathcal{A}_{a_L}\big)\,|a_1\cdots a_L\rangle\;, \qquad \mathcal{A}_a^{n,n'} = \frac{\lambda_{n'}}{N\sqrt{N}}\omega^{a(n-n')}\;. \tag{33}$$

It is straightforward to obtain the entanglement spectrum from this expression, as we discuss in the next section.

## 4. SPTs and RSPTs in the chiral-clock family

SPT phases are guaranteed to be stable only to perturbations preserving the protecting symmetries. One thus expects there to be a (non-symmetric) finite-depth local unitary transformation from a ground state with SPT order into a trivial product state [52]. The inverse of such a transformation is an SPT entangler [4–8, 53]. SPTs in clock models outside of the chiral family we consider have been studied previously [54, 55], and are interesting because of their relation to deconfined quantum critical points [28, 56–58]. For a review of SPT physics and for further references, see, for example, Refs. [2, 5, 59–61].

In Section 3.4 we showed how the ground state of $A_2$ indeed has such a form. Since it has SPT order for $N=2$, it is natural to hope that this property holds for all $N$. The situation, however, is subtler. An SPT with a protecting symmetry group $G = G_0$ or $G = G_0 \rtimes \mathbb{Z}_2^{\text{CPT}}$, where $G_0$ acts on-site, occurs when the group cohomology $H^2(G, U(1))$ is non-trivial [52, 61, 62]. We showed in Section 3.2 that $G = D_{2N} = \mathbb{Z}_N \rtimes \mathbb{Z}_2^{\text{CPT}}$ for our models. Since $H^2(D_{2N}, U(1)) = \mathbb{Z}_2$ for $N$ even and is otherwise trivial [63], we have the possibility of an SPT only for even $N$. Worth noting is that this protection for $N = 2$ differs from both the unitary $\mathbb{Z}_2 \times \mathbb{Z}_2$ and the anti-unitary $\mathbb{Z}_2 \times \mathbb{Z}_2^T$ symmetries that typically protect the SPT order of the $N = 2$ cluster model [7].

In this section, we probe deeper by analysing the properties of the MPS ground state (33) of $A_2$. In particular, we compute its entanglement spectrum and symmetry fractionalisation. We show that for even $N$ it is indeed an SPT state. For odd $N$ we find that it has behaviour reminiscent of an SPT, but without being as robust. Such phases were dubbed RSPTs in a closely related context [19], and we discuss how they arise here.

4.1. **No SPT for $A_2$ with odd $N$.** The distinction between odd and even $N$ is apparent in the entanglement spectrum of $|\psi_2\rangle$. Using $\lambda_n = \omega^{n/2}|\lambda_n|$ we write the matrix elements of $\mathcal{A}_j$ from (33) as

$$\mathcal{A}_a^{n,n'} = \Gamma_j^{n,n'}\Lambda_{n'}\;, \qquad \Gamma_a^{n,n'} \equiv N^{-\frac{1}{2}}\,\omega^{a(n-n')}\omega^{n'/2}\;, \qquad \Lambda_{n'} = N^{-1}\,|\lambda_{n'}|. \tag{34}$$

This MPS is in canonical form [64, 65] because the transfer matrix $T^{n,p;n',p'} = \sum_a \mathcal{A}_a^{n,n'}\,\overline{\mathcal{A}}_a^{p,p'}$ has dominant right (left) eigenvector $\delta_{n',p'}$ ($\Lambda_n^2\delta_{n,p}$) with eigenvalue 1. The entanglement spectrum for a bipartition of an open chain is then $\{\Lambda_n^2\}$, where

$$\Lambda_n^{-1} = N\sin\left(\frac{(2n+1)\pi}{2N}\right) \qquad n = 0, 1, \ldots N-1. \tag{35}$$

A necessary but not sufficient condition for SPT order is an exact degeneracy in the entanglement spectrum [20, 66, 67]. Since the argument of the sine in (35) is symmetric about $\pi/2$, there is indeed a two-fold degeneracy throughout the spectrum for $N$ even. We show in Section 4.2 that this exact degeneracy is a consequence of a projective representation of $D_{2N}$, implying SPT order.

For $N$ odd, however, the single non-degenerate Schmidt value $\Lambda_{(N-1)/2} = N^{-1}$ means that there is no non-trivial projective representation in the ground state of $A_2$. This observation is consistent with the lack of $D_{2N} = \mathbb{Z}_N \rtimes \mathbb{Z}_2^{\text{CPT}}$ SPT order for $N$ odd [63]. Moreover, it implies a stronger statement: for $N$ odd, $A_2$ cannot describe a non-trivial SPT phase, even if we have missed some symmetries. However, as only the lowest Schmidt value for $N$ odd is non-degenerate, we expect that some of the physics is independent of $N$. We show next that symmetry fractionalisation indeed holds for all $A_2$.

4.2. **Symmetry fractionalisation.** Symmetry fractionalisation in the ground state is characteristic of SPT order. In the MPS picture, this is particularly clear: if two sets of matrices $\mathcal{A}_a$ and $\mathcal{B}_a$ represent the same state, then $\mathcal{A}_a = e^{i\varphi} M \mathcal{B}_a M^{-1}$ for some phase $\varphi$ and invertible matrix $M$ on the bonds [68]. For an SPT with a unitary on-site symmetry group $G$ that preserves the ground state, applying a symmetry to the physical index gives us an equivalent state, and the corresponding $M$ are unitary and form a projective representation of $G$ [69]. The cohomology class of the representation classifies the SPT order [20,52,61–63]. In particular, one can implement symmetries in terms of such matrices $U(g)$ for $g \in G$ that satisfy [65]

$$U(g)\Lambda = \Lambda U(g) , \qquad \sum_{b=0}^{N-1} u(g)_{a,b}\, \widetilde{\Gamma}_b = e^{i\varphi(g)} U(g)\, \Gamma_a\, U(g)^\dagger , \tag{36}$$

where $\widetilde{\Gamma} = \Gamma$ for on-site global symmetries. When $g$ implements lattice inversion, we have $\widetilde{\Gamma} = \Gamma^T$, for the anti-unitary time-reversal $\widetilde{\Gamma} = \overline{\Gamma}$, and so for our combined CPT symmetry $\widetilde{\Gamma} = \Gamma^\dagger$.

The non-trivial symmetry fractionalisation for $A_2$ with even $N$ follows from the $\mathbb{Z}_2 \times \mathbb{Z}_2^{\text{CPT}}$ subgroup of our $D_{2N}$ symmetry. The $\mathbb{Z}_2$ symmetry from $X^{N/2}$ acting on a given site is implemented by matrices on the adjacent bonds by

$$\sum_{b=0}^{N-1} X_{a,b}^{N/2}\, \Gamma_b = Z^{N/2}\, \Gamma_a\, Z^{N/2} , \tag{37}$$

where the $X_{a,b}$ are the matrix elements of the operator $X$. For the $\mathbb{Z}_2^{\text{CPT}}$ symmetry,

$$\sum_{b=0}^{N-1} C_{ab} \left(\Gamma_b\right)^\dagger = e^{-2i\varphi}\left(e^{i\varphi}\sqrt{Z}V\right)\Gamma_j\left(e^{-i\varphi}V\sqrt{Z}^\dagger\right) \tag{38}$$

where $C_{ab}$ are the matrix elements of the single-site charge-conjugation operator given in (21), and

$$\sqrt{Z} = \sum_{j=0}^{N-1} \omega^{j/2}\, |j\rangle\, \langle j| , \qquad V = V^\dagger = \sum_{j=0}^{N-1} |j\rangle\, \langle N-1-j| , \qquad e^{i\varphi} = \omega^{-\frac{N-1}{4}} , \tag{39}$$

where the bras and kets here are for the bond states. Note that this choice of $V$ commutes with the $\Lambda$ matrix, and for both generators we have used the freedom to fix the phases to make $U(g)^2 = 1$. (The overall phase $e^{-2i\varphi}$ does not enter into the representation matrices.)

The non-trivial projective representation of $\mathbb{Z}_2 \times \mathbb{Z}_2^{\text{CPT}}$ on the bonds follows since the representation of these two generators does not commute:

$$\left(e^{i\varphi}\sqrt{Z}V\right) Z^{N/2} = -Z^{N/2}\left(e^{i\varphi}\sqrt{Z}V\right) . \tag{40}$$

The two-dimensional irreducible projective representations of the symmetry group on the bonds requires the observed two-fold degeneracy in the entanglement spectrum. These properties are stable away from the fixed point, and cannot change without a bulk phase transition. Since this SPT order is protected by a subgroup $\mathbb{Z}_2 \times \mathbb{Z}_2^{\text{CPT}} \leq D_{2N}$, the SPT phase of $A_2$ remains stable under any perturbations preserving the subgroup, even if they break the full $D_{2N}$.

Despite not having SPT order, we still have symmetry fractionalisation in the ground state of $A_2$ for odd $N$. Our analysis of the $\mathbb{Z}_2^{\text{CPT}}$ generator in Eq. (38) applies for odd $N$. For all $N$, the full $\mathbb{Z}_N$ symmetry is implemented by

$$\sum_{b=0}^{N-1} X_{a,b}\, \Gamma_b = Z^\dagger \Gamma_a Z, \tag{41}$$

generalising (37). For $N = 2p+1$ this action gives $p$ irreducible two-dimensional linear representations of $D_N$, each of which acts on the basis $\{|b\rangle, |N-1-b\rangle\}$ as

$$r = \begin{pmatrix} \omega^b & 0 \\ 0 & \omega^{N-1-b} \end{pmatrix} \qquad s = \begin{pmatrix} 0 & e^{i\varphi}\omega^{\frac{N-1-b}{2}} \\ e^{i\varphi}\omega^{\frac{b}{2}} & 0 \end{pmatrix} \tag{42}$$

for $b = 0, \ldots, p-1$. A single one-dimensional representation acts on $|p\rangle$ as $r = \omega^p$ and $s = e^{i\varphi}\omega^{p/2}$.

The singlet occurs for the space with the smallest Schmidt value. All of the others, including the dominant ones, form two-dimensional irreducible representations of the non-abelian $D_{2N}$ symmetry group. This dimension of course cannot change continuously, so for small enough symmetry-preserving perturbations it cannot change without the system undergoing some sort of transition. This property therefore implies local stability of the phase. A phase with such behavior was dubbed a "representation SPT" (RSPT) [19]. Similar physics appears in other settings, including AKLT chains with even spin [70], quotient symmetry-protected topological order [6] and boundary-obstructed topological phases [71]. A transition out of an SPT phase must be a bulk one, as the representations are projective. The protection for the RSPT, however, is not as strong. A sufficiently large perturbation can change the dimension of the dominant Schmidt value without encountering a bulk phase transition, as shown in [21] for the model of [19].

Thus for odd $N$ the ground state of $A_2$ can be deformed to that of $A_0$ without encountering a bulk phase transition or breaking the $D_{2N}$ symmetry, as one expects in the absence of non-trivial SPT order. Before this transition, however, the phase exhibits SPT-like physical properties such as symmetry fractionalisation.

4.3. **String order.** String order parameters can be used to identify an SPT phase without explicitly calculating the symmetry fractionalisation considered above [65]. Indeed, we will use them to this effect in Section 5.3. In this section we identify the relevant string order parameter for the symetry group $D_{2N} = \mathbb{Z}_N \rtimes \mathbb{Z}_2^{\mathrm{CPT}}$.

As above, there is a key distinction between even and odd $N$. Namely, the $\mathbb{Z}_N$ symmetry of the chiral-clock family possesses a $\mathbb{Z}_2$ subgroup for even $N$. We then can define a $\mathbb{Z}_2$ string operator using the "disorder" operator

$$\mu_k = \prod_{j=1}^{k-1} X_j^{N/2} , \qquad \left(\mu_k\right)^2 = 1, \qquad (43)$$

familiar from the Ising chain [72]. The limiting two-point function $\lim_{M,L\to\infty}\langle\mu_k\mu_{k+M}\rangle$ has a non-zero value only in the trivial phase. More generally we can dress $\mu_k$ with a local end-point operator $\mathcal{O}_k$ (which is supported on some finite region to the right of site $k-1$). The key idea is that we will see long-range order in $\langle\mu_1\mathcal{O}_1\ \mu_k\mathcal{O}_k\rangle$ only when $\mathcal{O}_k$ has symmetry properties that are consistent with the SPT phase [65]. This notion can be generalised also to critical points and gapless phases [6, 53, 73],

In our particular $D_{2N}$ setting, endpoint operators should have simple properties under conjugation by the CPT symmetry defined in Section 3.2. This symmetry is unusual since it combines an on-site unitary (charge conjugation), parity symmetry and anti-unitary time reversal. The ensuing complications require us to generalise the approach of [65] to such symmetries. In Appendix B we do so. Namely, we consider two-site hermitian endpoint operators $\mathcal{O}_{k,k+1}$ satisfying

$$C P \mathcal{K}\, \mathcal{O}_{k,k+1}\, C P \mathcal{K} = s_c\, \mathcal{O}_{\hat{k},\hat{k}+1} X_{\hat{k}}^{N/2} X_{\hat{k}+1}^{N/2} \qquad s_{\mathrm{c}} = \pm 1\ . \qquad (44)$$

There are two unusual aspects of this definition of the "charge" $s_c$, both due to applying the parity transformation $P$. First, the transformed operator is supported on the sites $\hat{k}, \hat{k}+1$, where $\widehat{k}$ is determined by which points are chosen to remain fixed under spatial inversion. Second, the charge is defined relative to multiplying by $X_{\hat{k}}^{N/2} X_{\hat{k}+1}^{N/2}$. The reason is that $P$ inverts the string $\mu_k$ as well as the end-point operator, and the former needs to be multiplied by the global $\mathbb{Z}_2 = \mu_L$ to return it to a left-pointing string. Since we consider operators $\mu_k\mathcal{O}_k$, the extra factors can be absorbed into the end-point operator by multiplying it by $X_{\hat{k}}^{N/2} X_{\hat{k}+1}^{N/2}$. An MPS-based derivation allowing for a general end-point operator supported on more than two sites is given in Appendix B.

The charge $s_c$ of the end-point of a string with long-range order reveals the SPT phase. In particular, for an end-point with $s_c = -1$, the asymptotic two-point function for $\mu_k\mathcal{O}_{k,k+1}$ is finite only in the SPT phase. (Recall there is only one non-trivial SPT phase for our symmetry group.) On the other hand, long-range order with $s_c = 1$ corresponds to the trivial phase.

The pivot procedure allows to find an end-point operator with $s_c = -1$ easily. Pivoting by the SPT entangler gives

$$U_1(\pi) X_k^{N/2} U_1(\pi) = \left( \sum_{m=1}^{N-1} \frac{1-(-1)^m}{1-\omega^m} Z_{k-1}^{-m} Z_k^m \right) X_k^{N/2} \left( \sum_{m=1}^{N-1} \frac{1-(-1)^m}{1-\omega^m} Z_k^{-m} Z_{k+1}^m \right) \qquad (45)$$

where the dressing term squares to one (see Appendix A). The string therefore pivots to

$$\mu_k \mu_{k+M} \to \left( \sum_{m=1}^{N-1} \frac{1-(-1)^m}{1-\omega^m} Z_{k-1}^{-m} Z_k^m \right) \mu_k \mu_{k+M} \left( \sum_{m=1}^{N-1} \frac{1-(-1)^m}{1-\omega^m} Z_{k+M-1}^{-m} Z_{k+M}^m \right). \qquad (46)$$

The end-point operators and the string overlap, and absorbing the ends of the former into the latter gives

$$\mathcal{O}_{k,k+1} = i X_k^{N/2} \left( \sum_{m=1}^{N-1} \frac{1-(-1)^m}{1-\omega^m} Z_k^{-m} Z_{k+1}^m \right) \quad \Longrightarrow \quad \mu_k \mu_{k+M} \to \mu_k \mathcal{O}_{k-1,k} \; \mu_{k+M-1} \mathcal{O}_{k+M-1,k+M} \; . \qquad (47)$$

The factor of $i$ ensures $\mathcal{O}_{k,k+1}$ is Hermitian and that it transforms under CPT as in (44) with $s_c = -1$. For $N=2$ it reduces to $i\sigma_k^x \sigma_k^z \sigma_{k+1}^z = \sigma_k^y \sigma_{k+1}^z$, the usual cluster-state end-point operator [42].

The ground state of $A_0$ is a product state and hence obviously in a trivial phase. Indeed, the disorder operator itself has long-range order:

$$\left\langle v_1^{(0)} v_2^{(0)} \cdots v_L^{(0)} \middle| \mu_k \, \mu_{k+M} \middle| v_1^{(0)} v_2^{(0)} \cdots v_L^{(0)} \right\rangle = 1 \; . \qquad (48)$$

We then exploit $\mu_{k+1} = \mu_k X_k^{N/2}$ and note that $X_k^{N/2}$ transforms under CPT as in (44) with $s_c = 1$. Thus our approach reproduces the triviality of the ground state of $A_0$.

Pivoting using (32) and (46) we then see immediately that (48) requires

$$\langle \psi_2 | \, \mathcal{O}_{k-1,k} \, \mu_{k+1} \, \mu_{k+M-1} \, \mathcal{O}_{k+M-1,k+M} \, | \psi_2 \rangle = 1 \; . \qquad (49)$$

The ground state of $A_2$ therefore has long-range order for a string operator with end-point operator that obeys (44) with $s_c = -1$. We conclude that $A_2$ is in a non-trivial SPT phase, distinct from $A_0$, protected by $\mathbb{Z}_2 \times \mathbb{Z}_2^{\mathrm{CPT}} \leq D_{2N}$ for all even values of $N$. We thus recover the result found using symmetry fractionalisation in the preceding Section 4.2. The approach here emphasises the role of the $\mathbb{Z}_2$ symmetry present only at even $N$. Moreover, computing the string order via the operators in (49) provides a useful diagnostic tool for detecting a non-trivial phase away from the special points with an exact MPS ground state.

While the choice of end-point operator in Eq. (47) appears naturally by applying the SPT entangler to the disorder operator, there exist simpler end-point operators with the correct charge that may be useful in some situations:

$$\mathcal{O}'_{k,k+1} = \begin{cases} i X_k^{2n-1} Z_k^{2n-1} Z_{k+1}^{2n-1} & N = 4n-2 \\ i X_k^{2k} \left( Z_k^{-1} Z_{k+1} + Z_k Z_{k+1}^{-1} \right) & N = 4n \end{cases} . \qquad (50)$$

## 5. How the phases fit together

We have discussed in depth the three Hamiltonians $A_0$, $A_1$ and $A_2$ for the chiral-clock family. They respectively have no order, spontaneous symmetry breaking, and (R)SPT order. To probe the physics further, we combine them and analyse the Hamiltonian

$$H(\alpha, \beta, \gamma) = \alpha A_0 + \beta A_1 + \gamma A_2. \qquad (51)$$

The Hamiltonian $H(\alpha, \beta, \gamma)$ is integrable, as noted above in (18). However, deriving properties for $N > 2$ is rather difficult not only because of the interactions, but also due to the presence of level-crossing transitions in the ground state [31]. Nonetheless along certain lines we utilise and obtain analytic results. We also utilise density matrix renormalisation group (DMRG) [74, 75] numerics to understand the phase diagram for $N = 3$ and $N = 4$. We find that all these orderings extend away

from these special solvable points, occupying regions of the phase diagram. For $N = 2$, the transitions between phases are direct, but for larger $N$ intermediate gapless regions typically appear.

5.1. **Special lines.** Three special lines of couplings give useful insight into the phase diagram.

5.1.1. *The Onsager-integrable chiral Potts line.* The line $A_1 + \lambda A_0$ is the canonical Onsager-integrable chiral Potts chain. Much is known from extensive work some decades ago [13, 16, 18, 30, 31, 48, 49, 76, 77], but many puzzles remain. The main difficulty is that the lack of a $U(1)$ symmetry makes a traditional Bethe-ansatz analysis impractical. The ground-state phase diagram for $N = 3$ was analysed carefully in [31, 76, 78]. Along with the symmetry-breaking phase at $\lambda = 0$ and trivial phase for large $\lambda$, for $\lambda > 0$ there are two intermediate gapless phases for $\lambda_c < \lambda < 1$ and $1 < \lambda < 1/\lambda_c$, where $\lambda_c \simeq 0.901$ [31]. More recently, DMRG calculations at $\lambda = 1$ [79] found oscillations in the scaling of entanglement entropy for open boundaries. This behaviour is consistent with previous results for Lifshitz transitions [80]. The case of $\lambda < 0$ is equivalent, as follows from the results of Section 3.3 .

Moreover, considering the energy of unit-charge excitations, long-range spin order occurs for all $\lambda < 1$ [31], including the intermediate gapless phase $\lambda_c < \lambda < 1$. We have not been able to observe the latter feature in our numerical studies below, and think that this long-range order in the gapless region is worth a deeper investigation. This intermediate region is further studied in Ref. [81], where scaling exponents for the order parameter are found; note that this gapless region is not described by a conformal field theory as the left- and right-moving excitations have different velocities [30, 81, 82]. These results can be summarised in the phase diagram

$$\text{(52)}$$

For $N > 3$ we are not aware of similar results for the phase diagram, although there are general formulae for structure of the spectrum [13, 17, 76, 77] and both numerical and analytic studies of the spectrum for small system sizes [31, 77]. For general $N$ the ground state in the zero-momentum sector has a transition at $\lambda = 1$ [77, 82]. This is not necessarily the ground state of the Hamiltonian because there may be a level crossing to a different momentum sector. In such a case, translation symmetry breaking [83] and/or an intermediate gapless phase or phases must occur. Our numerical studies described below in Section 5.3 indicate a similar structure for $N = 4$, including a first-order transition into an intermediate gapless phase.

A remarkable formula for the symmetry-breaking order parameter in the ferromagnetic phase was conjectured in [77] and proved (subject to certain analyticity assumptions) in [84, 85]. It is

$$\lim_{M,L\to\infty} \left\langle Z_1^{-k} Z_M^k \right\rangle = (1 - \lambda^2)^{\frac{k(N-k)}{N^2}} \qquad |\lambda| < \lambda_0, \tag{53}$$

where $\lambda_0$ indicates the first ground state phase transition we encounter beyond $\lambda = 0$ (moreover, for $N = 3$ we have a non-zero expectation for all $|\lambda| < 1$, based on the analysis of Ref. [31]). Duality yields

$$\lim_{M,L\to\infty} \left\langle \prod_{j=1}^{M} X_j^k \right\rangle = (1 - \lambda^{-2})^{\frac{k(N-k)}{N^2}} \qquad |\lambda| > \lambda_0^{-1}. \tag{54}$$

The (trivial) string-order parameter thus takes the $N$-independent value

$$\lim_{M,L\to\infty} \left\langle \mu_1 \mu_M \right\rangle = (1 - \lambda^{-2})^{\frac{1}{4}} \qquad |\lambda| > \lambda_0^{-1}. \tag{55}$$

5.1.2. *The line $A_1 + \lambda A_2$.* Pivoting the preceding Hamiltonian with $U_1(\pi)$ means that the Hamiltonian $A_1 + \lambda A_2$ is unitarily equivalent. The transitions must therefore occur at the same values of $\lambda_c$. However, the physical interpretation of the phases is rather different. We saw already that $A_2$ possesses (R)SPT

order. Transforming the trivial string order from (55) immediately shows the SPT order exists away from $A_2$. Namely, using (46) yields exact topological string order at even $N$:

$$\lim_{M,L\to\infty} \langle \mu_1 \mathcal{O}_{1,2} \, \mu_M \mathcal{O}_{M,M+1} \rangle = (1 - \lambda^{-2})^{\frac{1}{4}} \qquad |\lambda| > \lambda_0^{-1}. \tag{56}$$

The SPT order therefore persists at least until $\lambda = \lambda_c^{-1}$ (as it must on general stability grounds), and likely all the way until $\lambda = 1$, as summarized in the diagram

$$\tag{57}$$

Subtleties with SPT physics arise in gapless models [6], but resolving them requires a deeper understanding of the nature of the gapless phase realised here. Moreover, we cannot prove that the RSPT phase at odd $N$ persists, even in the gapped region, but we expect that the dominant Schmidt value remains doubly degenerate throughout. Below we give numerics in support of this contention.

5.1.3. *The $U(1)$ line and the exact ferromagnetic ground state.* Setting $\alpha = \gamma$ in (51) yields a rather special line of Hamiltonians. Namely, it follows immediately from (4) that $A_1$ commutes with $A_0 + A_2$:

$$\big[A_1, \alpha A_0 + \beta A_1 + \alpha A_2 \big] = 0 \ . \tag{58}$$

Since $N A_1$ has integer eigenvalues, it generates a $U(1)$ symmetry along this line. An explicit expression for the KW dual Hamiltonian in terms of the usual $SU(2)$ operators $S^{\pm}, S^z$ can be found in [34].

This $U(1)$ symmetry allows the coordinate Bethe ansatz to be used, as described in depth in [34] for a closely related $U(1)$-invariant model. Acting with the Onsager generators turns out to correspond to adding or removing "exact strings" within the Bethe ansatz. A conjecture was made there that the ground state of the model $A_1 + A_{-1}$ (the dual of $A_0 + A_2$) is comprised solely of such exact strings. Our detailed calculations show however that this property holds true only for $L \leq 12$. Thus the analysis, while tractable, is still rather difficult. We defer a full accounting to a separate paper [33].

However, the $U(1)$ symmetry does result in an exact ground state over a range of $\alpha/\beta$ when $\alpha = \gamma$. When $H = A_1$, the $N$ ground states are given by all spins equal in the $Z$-basis, as shown in (29). Each such state is annihilated by $A_0 + A_2$. Moving away from $A_1$ by allowing $\alpha = \gamma \neq 0$, the different ground states do not mix in perturbation theory until order $L$, and the exact ferromagnetic ground states persist until the $\alpha = \gamma$ is of order $\beta$. For $N = 2$, this transition occurs exactly at $\alpha = \gamma = \beta/2$. This value is recovered by a simple first-order perturbation theory in the one-particle sector [86], giving a transition at $\alpha/\beta = \gamma/\beta = \sin(\pi/N)/2$. For $N = 3, 4$, and for $\alpha = \gamma = 1/2$, this predicts $\beta \simeq 1.15, 1.41$ respectively. These values are consistent with the numerics in Fig. 4, though the transition appears to occur for a larger value of $\beta$ in the $N = 4$ case.

5.2. **Phase diagram for $N = 2$.** We first consider the full phase diagram in the $N = 2$ free-fermion case, where it is known exactly. This case is KW dual to the usual quantum XY model [87], and so phase transitions occur at the same places. The results are displayed in Fig. 3(a), with the trivial, SPT and ferromagnetic phases readily apparent.

The line from Section 5.1.1 with $\gamma = 0$ corresponds at $N = 2$ to the usual transverse-field Ising model, while pivoting with $A_1$ yields the line with $\alpha = 0$ described in (5.1.2). Onsager's results show that these models have Ising critical points at $\alpha = \beta$ and $\gamma = \beta$ respectively. The latter criticality is enriched by $\mathbb{Z}_2 \times \mathbb{Z}_2^{\mathrm{CPT}}$ [6], as confirmed by our analysis of the string order in Section 4.3 for all even $N$. Since the disorder operator has charge $s_c = 1$ and $s_c = -1$ under CPT for trivial and SPT phases respectively, the corresponding critical points are labelled Ising$^{\pm}$.

The transition between the trivial and SPT phases along the $\beta = 0$ line occurs at the $U(1)$ invariant value $\alpha = \gamma$ for $N = 2$. Its continuum limit is described by a single free-boson field theory, a conformal field theory with central charge $c = 1$. The type of criticality is invariant (and the $c = 1$ remains at the free-fermion radius) up to the multicritical point at $\alpha = \gamma = \beta/2$. The multicritical point has

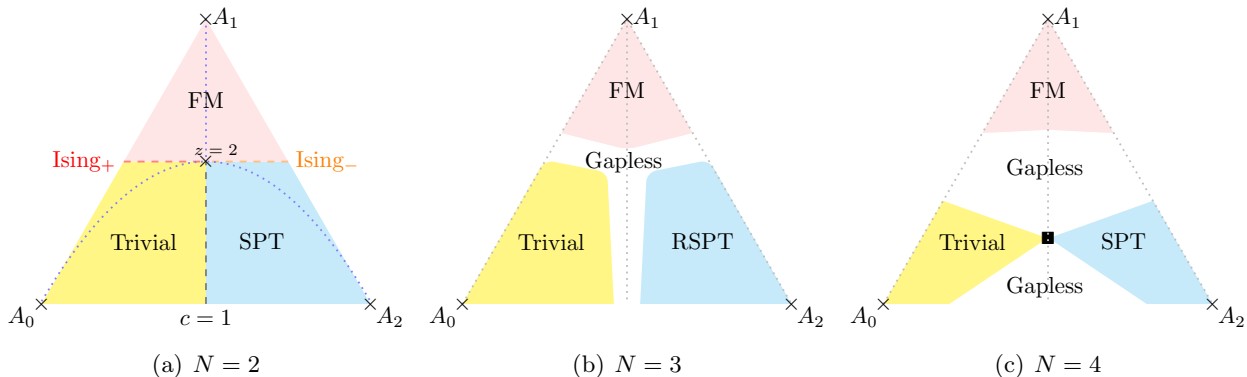

(a) $N = 2$          (b) $N = 3$          (c) $N = 4$

FIGURE 3. Schematic phase diagrams for $N = 2, 3, 4$ for the Hamiltonian $H(\alpha, \beta, \gamma)$ parameterised as $\alpha + \beta + \gamma = 1$. FM indicates the ferromagnetic phase. The $N = 2$ phase diagram is known exactly and features direct transitions. An exact MPS ground state occurs on the dotted line. For $N = 3, 4$ the transitions typically spread out into gapless regions. Within the achieved numerical resolution, we cannot ascertain whether we have a narrow gapless region or a direct transition between the trivial and SPT phases for a certain range along the $\alpha = \gamma$ line for $N = 4$ (indicated by ■). The grey dotted lines are discussed in Section 5.1.

dynamical critical exponent $z = 2$, and the charge of the disorder operator changes sign along the Ising CFT line here. This point also has an exact MPS ground state, as does the model along the dotted line [88–91] with couplings $\alpha = (1 - \lambda)^2, \beta = 2\lambda(1 - \lambda), \gamma = \lambda^2$. This line is dual to the disorder line in the XY model [92–94].

5.3. **Phase diagram for $N = 3, 4$.** To determine the full phase diagrams $H(\alpha, \beta, \gamma)$ for $N = 3, 4$, we need to distinguish the three phases: trivial, symmetry breaking and (R)SPT dominated by $A_0, A_1, A_2$ respectively. Here we use the density matrix renormalization group (DMRG) [95] to go beyond the above analytic results. All following numerical calculations were performed using the ITensor library [74, 75] for finite systems with open boundary conditions. We summarise our results in Fig. 3, showing that the critical lines seen in the $N = 2$ case broaden out to gapless regions. A key result is that the RSPT order at $N = 3$ remains throughout a region.

We use a variety of probes to determine the phase diagrams. For example, we give results from entanglement entropy in Fig. 4. The probes are:

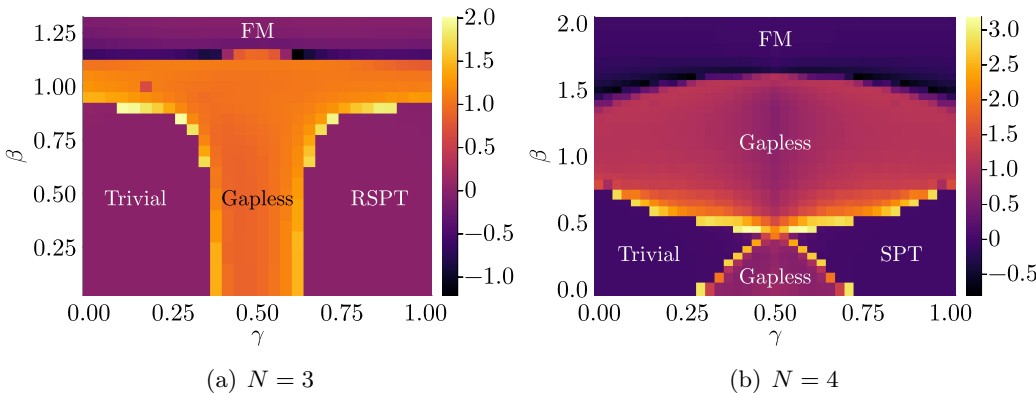

(a) $N = 3$          (b) $N = 4$

FIGURE 4. The effective central charge $c$ for $H(\alpha, \beta, \gamma)$ with $\alpha = 1 - \gamma$ for (a) $N = 3$ and $L = 100$, (b) $N = 4$ and $L = 40$. The value is extracted by fitting the entanglement entropy to the CFT formula Eq. (63). A zero value indicates an area-law ground state. Values of $c$ at the boundary of the gapless region are not meaningful. A unitary transformation relates Hamiltonians with $\gamma$ and $1 - \gamma$, and so the data for $N = 4$ for $\gamma > 0.5$ is that for $\gamma \leq 0.5$.

5.3.1. *Local and string order parameters:* A non-vanishing value two-point correlation function

$$\mathcal{O}_Z = \left| \langle Z_j Z_k^{-1} \rangle \right| \tag{59}$$

at large $|j - k|$ indicates ferromagnetic order. As seen in Fig. 5(b) and (d), such order occurs at large $\beta$ where $A_1$ dominates the Hamiltonian.

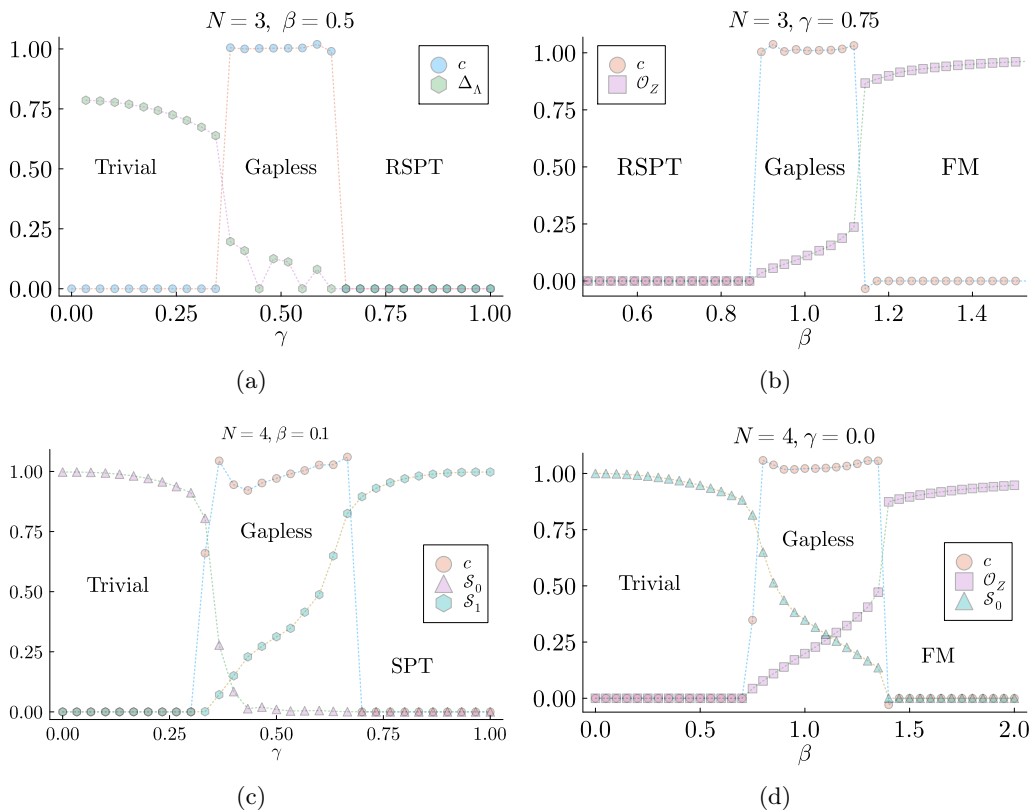

FIGURE 5. DMRG calculations for the $N = 3$ (a,b) and $N = 4$ (c,d) versions of the Hamiltonian (51) with $\alpha = 1 - \gamma$ and with system sizes $L = 200$ and $L = 100$ respectively. $\mathcal{O}_Z$ is the local order parameter shown in Eq. (59), $\mathcal{S}_0$ and $\mathcal{S}_1$ are the trivial and non-trivial string order parameters defined in Eqs. (60) and (61) respectively, while $c$ comes from fitting the entanglement entropy to Eq. (63). The Schmidt value $\Lambda_\alpha$ comes from the mid-chain bipartite Schmidt decomposition of the ground state, and the difference between the largest two values $\Delta_\Lambda = \Lambda_1^2 - \Lambda_2^2$. Dotted lines connecting data points are provided as a guide to the eye.

The string operators discussed in Section 4.3 provide a convenient way to distinguish the SPT phase and trivial phases at even $N$. The trivial phase at even $N$ is detected by the two-point function of disorder operators, namely

$$\mathcal{S}_0 = |\langle \mu_j \mu_k \rangle|. \tag{60}$$

The SPT phase, on the other hand, is detected by pivoting Eq. (60) using the SPT entangler as discussed in Section 4.3. For $N = 4$, we show in Appendix B.2, that the relevant string order (47) takes on the form

$$\mathcal{S}_1 = \left| \left\langle \frac{i}{2} \left( S^{-1,-1} - S_{1,1} \right) + S^{-1,1} \right\rangle \right|, \qquad S^{a,b} \equiv Z_{j-1}^a Z_j^{-a} \mu_j \mu_{k+1} Z_k^b Z_{k+1}^{-b} \tag{61}$$

Non-vanishing values of these two-point functions Equations (60) and (61) for large values of $|j - k|$ are good order parameters for the trivial and SPT phases respectively. We plot these values in Fig. 5(c,d) for $j = L/4$ and $k = 3L/4$, indicating the presence of these phases when $A_0$ and $A_2$ respectively dominate.

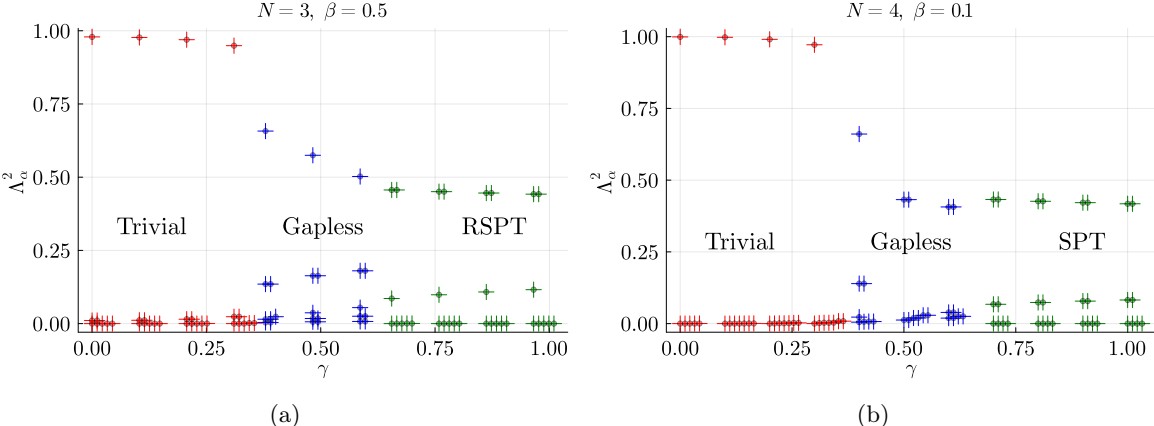

FIGURE 6. The entanglement spectrum for the ground state of (51) with $\alpha = 1 - \gamma$ and system size $L = 200$. All levels are doubly degenerate in the SPT phase, while both degenerate and non-degenerate values occur in the RSPT and trivial phases. The latter two phases are distinguished by the dominant entanglement level, which is unique for the trivial phase but degenerate for the RSPT.

5.3.2. *Entanglement spectrum.* The bipartite entanglement spectrum of the ground states provides a useful probe for detecting the SPT and RSPT phases. The values are $\Lambda_\alpha^2$, where the $\Lambda_\alpha$ are the usual Schmidt coefficients of the ground state [20, 65, 66]. The entanglement spectrum of SPT phases is characterized by the presence of robust degeneracies for all $\Lambda_\alpha$ and the total absence of non-degenerate values, shown in Fig. 6(b). The trivial and RSPT phases contain non-degenerate entanglement levels along with degenerate ones. These two phases are distinguished by the nature of the dominant Schmidt value. In the trivial phase it is unique, whereas in the RSPT phase it is degenerate, as shown in Fig. 6(a). These results confirm the stability of our analytic results in Section 4.1 away from the fixed point model $A_2$ for both even and odd $N$. By tracking the gap $\Delta_\Lambda = \Lambda_1^2 - \Lambda_2^2$ between the leading entanglement values, we can distinguish the trivial from the RSPT. As apparent in Fig. 5, $\Delta_\Lambda$ vanishes for the former but not the latter.

In a general context, the entanglement spectrum is known to be an indirect probe of the edge modes in a topological phase [96]. Similarly, the degeneracy of the largest entanglement levels in the RSPT is expected to result in parametrically stable edge modes, despite lacking in topological protection [19, 21, 26, 27]. A simple pivoting calculation reveals that the fixed-point $A_2$ with open boundaries has $N^2$ ground states. However, the relation between entanglement spectrum and edge modes is known to break down when parity symmetry is involved [70], and indeed one can find symmetric boundary terms that gap out these edge modes for any non-zero coupling. Thus, the entanglement spectrum is the most relevant probe of the (R)SPT physics.

5.3.3. *Entanglement entropy.* The von Neumann entanglement entropy provides a good way to distinguish the gapless phases from the gapped ones. It is defined as

$$S(l_A) = \text{tr}(\rho_A \log \rho_A) \ , \tag{62}$$

where $\rho_A$ is the reduced density matrix with support on the Hilbert space $A$, taken to be a contiguous interval length $l$ on the chain. For ground states of one-dimensional gapped phases, $S(l) \sim \text{const}$, an area law. Critical gapless phases that are described by a 1+1-dimensional conformal field theory obey a universal form [97] of entanglement scaling in the large-$L$ limit. For a finite system of size $L$ with open boundary conditions,

$$S(l) = \frac{c}{6} \log \left( \frac{L}{\pi} \sin \left( \frac{\pi l}{L} \right) \right) + \text{const} \ , \tag{63}$$

where $c$ is the central charge that characterizes the CFT. Entanglement entropy obeying the form Eq. (63) thus indicates gapless behaviour, with $c \neq 0$ providing a measure of entanglement.

As seen in Fig. 5, this fit can be used effectively to distinguish the gapless phases from the surrounding gapped ones. We apply this procedure to locate the gapless regions in Figs. 3 and 4. We find that the data fits well to Eq. (63) with $c = 1$, as shown in Fig. 7, suggesting that the gapless states are described by a compact boson CFT or its orbifold [98]. However, as shown in Fig. 7(b), we observe oscillations in certain ranges of parameter values. As discussed in Section 5.1.1, these are also consistent with Lifshitz transitions [80] and have been observed in numerical investigations of similar models [79]. Similar results are seen for the gapless states appearing in the phase diagram of the $N = 4$ Hamiltonian. The precise nature of the gapless states appearing in our phase diagrams is an interesting question, which we leave for upcoming work.

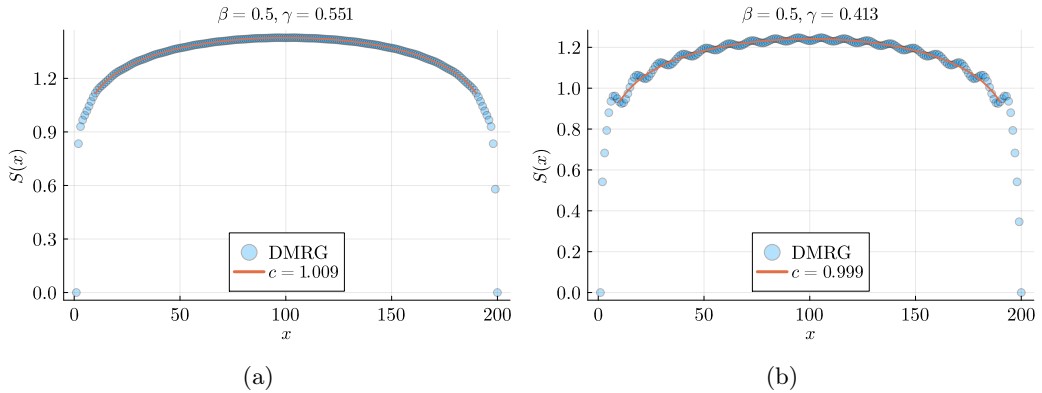

FIGURE 7. Von Neumann entanglement entropy computed for representative gapless ground states of the $N = 3$ Hamiltonian (51) with $\alpha = 1 - \gamma$ and system size $L = 200$, (datapoints) compared with the best fit to the CFT formula Eq. (63). Both have a best fit of $c \approx 1$ (solid line), with oscillations observed for some parameters (b) but absent for others (a).

## 6. Outlook

In this paper we showed how the Onsager algebra naturally gives rise to a pivot procedure useful for constructing SPT phases. We applied this result to a family of $N$-state chiral clock Hamiltonians constructed with this algebra. We found an SPT phase for even $N$ and an RSPT phase for odd $N$, protected by the dihedral group $D_{2N}$ comprised of the clock and CPT symmetries.

We analysed in depth the Hamiltonian $A_2$, the $N$-state analog of the cluster-model SPT. We found an analytic expression for the entanglement spectrum in its MPS ground state, and showed it has dominant, degenerate Schmidt values in the entanglement spectrum. For even $N$, the ground state has non-trivial SPT order characterised by a projective representation of $D_{2N}$ on the bond Hilbert space. We showed that this SPT phase can be detected by a string order parameter with an end-point charged under $\mathbb{Z}_2^{\mathrm{CPT}}$. For odd $N$, however, every Schmidt value of the ground state of $A_2$ is degenerate apart from the smallest one. This entanglement spectrum is inconsistent with any projective representation on the bonds, and thus corresponds to a trivial SPT phase (for any protecting symmetry group). However, the system has dominant degenerate Schmidt values corresponding to higher-dimensional irreducible representations of $D_{2N}$ and we conclude that $A_2$ represents an RSPT phase for $D_{2N}$.

The phase diagram interpolating between $A_0$, $A_1$ and $A_2$ is rich, and based on analytic and numerical results we conjectured its form for $N = 3, 4$. For even $N$ the three fixed-point Hamiltonians represent distinct phases of matter, and thus are separated by bulk transitions. We demonstrated that the RSPT phase in the case $N = 3$ does extend away from the fixed-point $A_2$ by numerically calculating the entanglement gap using DMRG [74, 75]. We moreover see a critical phase separating this RSPT from the trivial phase, which is "unnecessary" from the SPT point of view [24–28].

A key topic for future work is to better understand the phase diagrams outlined in Section 5.3. Particularly interesting would be substructure within the gapless regions, and possible symmetry enrichment. Indeed, our numerical investigations leave open the possibility that the $N = 4$ phase diagram contains

two separate gapless regions meeting at a direct transition between the trivial and SPT phases [27,73]. In a companion work, we consider the $U(1)$ invariant line $A_0 + A_2 + hA_1$, along with the KW dual $A_1 + A_{-1} + hA_0$ (where the $U(1)$ symmetry is on-site), note that this line includes the possible direct transition. This symmetry lends itself to a coordinate Bethe Ansatz approach. We have left the wider phase diagram for the chiral Potts family $H = \sum t_\alpha A_\alpha$ unexplored. While there is only one non-trivial SPT phase for our symmetry group, we may see combined symmetry-breaking and SPT physics in the higher Hamiltonians [5]. Identifying the gapless regions in this larger phase diagram would also be interesting.

Our results are not restricted to the Onsager-integrable chiral-clock family we have studied. The $D_{2N}$ symmetry identified here occurs in more general chiral clock models. The $\mathbb{Z}_2^{\mathrm{CPT}}$ symmetry takes, for example, $\gamma X_n \to \overline{\gamma} X_n^\dagger$. It follows that any hermitian $H = \sum_j \sum_{m=1}^{N-1} \left( \gamma_m h_{2j}^m + \delta_m h_{2j+1}^m \right)$ has this symmetry, along with the standard clock symmetry. Hence, the SPT phase (and possibly the RSPT phase) will extend into the wider phase diagram of the chiral clock models [79].

A natural next step is to look for SPT physics and interesting phase diagrams in other families of spin chains that generate an Onsager algebra, such as those in Refs. [99–101]. Further examples can be found in the setting of generalised Onsager algebras; there we have multiple generators each satisfying a mutual Dolan-Grady relation [102,103]. A fundamentally different model to those considered in this paper is the "free fermions in disguise" Hamiltonian [104], which can be written as a sum of terms satisfying a generalised Onsager algebra [103]. It would be most interesting to uncover a new family in this class.

One final outstanding question is to find which subspace of the phase diagram has an exact MPS ground state, generalising the line in Fig. 3(a). The solution to this problem for $N = 2$ utilises imaginary time evolution with fixed-point Hamiltonians [90], and the Onsager algebra may allow for a generalisation of this approach.

**Acknowledgements** We are grateful to Murray Batchelor, Yuchi He, Max McGinley, Yuan Miao and Ryan Thorngren for helpful discussions and correspondence. We also thank Ruben Verresen and Nathanan Tantivasadakarn for sharing their insights and their related results [21]. This work was supported by the European Research Council under the European Union Horizon 2020 Research and Innovation Programme, Grant Agreement No. 804213- TMCS (A.P), and by EPSRC grant EP/S020527/1 (P.F.). Preliminary work was completed while N.G.J. held a Heilbronn Research Fellowship at the Mathematical Institute, University of Oxford.

## Appendix A. Derivation of the Hamiltonian $A_2$

In this appendix we derive the closed-form expression for $A_2$ given in the main text (17). There are two paths one can take

$$A_2 = U_1 A_0 U_1^\dagger \qquad \text{and} \qquad A_2 = A_0 + \frac{1}{8} \left[ A_1, [A_0, A_1] \right] . \tag{A1}$$

We give both the pivot and commutator approaches for the Kramers-Wannier dual operator $A_{-1}$, and then connect the two to reach our preferred representation for $A_2$.

A.1. **Pivoting $A_1$ with $A_0$ to find $A_{-1}$.** We first use the pivot procedure to find $A_{-1} = U_0(\pi) A_1 U_0(\pi)^\dagger$. We can write $U_0(\pi)$ as a product of single-site terms, so that its action on the single-site operator $Z$ is

$$\widehat{Z} = e^{i\frac{\pi}{N} \sum_{m=1}^{N-1} \alpha_m X^m} Z \, e^{-i\frac{\pi}{N} \sum_{m=1}^{N-1} \alpha_m X^m}. \tag{A2}$$

In the $X$ basis, $X = \sum_a \omega^{-a} \left| v^{(a)} \right\rangle \left\langle v^{(a)} \right|$ and $Z = \sum_a \left| v^{(a-1)} \right\rangle \left\langle v^{(a)} \right|$, so

$$\widehat{Z} = \sum_{a=0}^{N-1} e^{i\frac{\pi}{N} \sum_{m=1}^{N-1} (\alpha_m \omega^{-m(a-1)} - \alpha_m \omega^{-ma})} \left| v^{(a-1)} \right\rangle \left\langle v^{(a)} \right|$$

$$= e^{i\frac{\pi}{N}} \sum_{a=0}^{N-1} (-1)^{\delta_{a,0}} \left| v^{(a-1)} \right\rangle \left\langle v^{(a)} \right| = e^{i\pi/N} Z \hat{\Phi}^{(0)}, \tag{A3}$$

where

$$\widehat{\Phi}_j^{(r)} \equiv \sum_a (-1)^{\delta_{r-a,0}} \left| v_j^{(a)} \right\rangle \left\langle v_j^{(a)} \right|, \qquad \widehat{\Phi}_j^{(r)} Z_j = Z_j \widehat{\Phi}_j^{(r+1)}. \tag{A4}$$

Obviously, $\left( \widehat{\Phi}_j^{(r)} \right)^2 = 1$ so that

$$U_0(\pi) Z_j^k U_0(\pi)^\dagger = \omega^{k/2} Z_j^k \prod_{r=0}^{k-1} \widehat{\Phi}_j^{(r)}, \tag{A5}$$

$$A_{-1} = U_0(\pi) A_1 U_0(\pi)^\dagger = -\frac{4}{N} \sum_{j=1}^{L} \sum_{m=1}^{N-1} \alpha_m \left( \prod_{r=0}^{m-1} \widehat{\Phi}_j^{(r)} \right) Z_j^{-m} Z_{j+1}^m \left( \prod_{r=0}^{m-1} \widehat{\Phi}_{j+1}^{(r)} \right). \tag{A6}$$

A.2. **Pivoting $A_0$ with $A_1$ to find $A_2$.** One can find $A_2$ by exchanging $A_0$ and $A_1$ in the preceding pivot, as follows from (11). The calculation then proceeds identically if one utilises the operators $h_k$ from (15). One then finds

$$A_2 = -\frac{4}{N} \sum_{j=1}^{L} \sum_{m=1}^{N-1} \alpha_m \left( \prod_{r=0}^{m-1} \Phi_{j-1,j}^{(r)} \right) X_j^m \left( \prod_{r=0}^{m-1} \Phi_{j,j+1}^{(r)} \right). \tag{A7}$$

where

$$\Phi_{j,j+1}^{(r)} = \sum_{a_1,a_2} (-1)^{\delta_{a_2-a_1+r,0}} \left| a_1 \right\rangle_j \left\langle a_1 \right|_j \left| a_2 \right\rangle_{j+1} \left\langle a_2 \right|_{j+1}. \tag{A8}$$

We show in Appendix A.4 how to rewrite this product of signs in terms of $Z_j$ operators.

A byproduct of this calculation is that $U_1(\pi) X_j U_1(\pi)^\dagger = \Phi_{j-1,j}^{(0)} X_j \Phi_{j,j+1}^{(0)}$. Since $X$ acts as a shift in the $Z$-basis, we have

$$\Phi_{j,j+1}^{(r)} X_j = X_j \Phi_{j,j+1}^{(r+1)}, \qquad \Phi_{j,j+1}^{(r)} X_{j+1} = X_{j+1} \Phi_{j,j+1}^{(r-1)}, \tag{A9}$$

yielding (45) and the resulting transformed string operator.

A.3. **Commutator calculation.** In this section we derive a closed-form expression for $A_{-1}$ using commutation relations directly.

Using the definitions of $A_0$ and $A_1$ from (15,16) gives

$$[A_0, A_1] = \left( \frac{4}{N} \right)^2 \sum_{j=1}^{L} \sum_{a,\hat{a}=1}^{N-1} \alpha_a \alpha_{\hat{a}} \left( 1 - \omega^{a\hat{a}} \right) \left( h_{2j-1}^a h_{2j}^{\hat{a}} - h_{2j}^{\hat{a}} h_{2j+1}^a \right). \tag{A10}$$

A useful identity proved below is

$$\sum_{a,b=1}^{N-1} \alpha_{a,\hat{a}} \alpha_{b,\hat{a}} h_{2j-1}^{a+b} = \hat{a}(N - \hat{a}) + (N - 2\hat{a}) \sum_{s=1}^{N-1} \alpha_{s,\hat{a}} h_{2j-1}^s, \tag{A11}$$

where we define $\alpha_{a,\hat{a}} = \alpha_a(1 - \omega^{a\hat{a}})$. Commuting $A_0$ with (A10) and using this identity yields

$$\left[ A_0, [A_0, A_1] \right] = -\left( \frac{4}{N} \right)^3 \sum_{j=1}^{L} \left( -2 \sum_{a,\hat{a},b=1}^{N-1} \alpha_{a,\hat{a}} \alpha_{\hat{a}} \alpha_{b,\hat{a}} h_{2j-1}^a h_{2j}^{\hat{a}} h_{2j+1}^b \right.$$
$$\left. + \sum_{a,\hat{a}=1}^{N-1} \alpha_{a,\hat{a}} \alpha_{\hat{a}} (N - 2\hat{a}) \left( h_{2j-1}^a h_{2j}^{\hat{a}} + h_{2j}^{\hat{a}} h_{2j+1}^a \right) + 2 \sum_{\hat{a}=1}^{N-1} \hat{a}(N - \hat{a}) \alpha_{\hat{a}} h_{2j}^{\hat{a}} \right). \tag{A12}$$

The Onsager algebra requires

$$A_{-1} = A_1 - \frac{1}{8} \left[ A_0, [A_0, A_1] \right], \tag{A13}$$

so that

$$
\begin{aligned}
A_{-1} = -\frac{4}{N^3} \sum_{j=1}^{L} \Bigg( & 4 \sum_{a,\hat{a},b=1}^{N-1} \alpha_a \alpha_{\hat{a}} \alpha_b \, (1 - \omega^{a\hat{a}})(1 - \omega^{b\hat{a}}) \, h_{2j-1}^a h_{2j}^{\hat{a}} h_{2j+1}^b \\
& - 2 \sum_{a,\hat{a}=1}^{N-1} \alpha_a \alpha_{\hat{a}} \, (N - 2\hat{a})(1 - \omega^{a\hat{a}}) \left( h_{2j-1}^a h_{2j}^{\hat{a}} + h_{2j}^{\hat{a}} h_{2j+1}^a \right) + \sum_{\hat{a}=1}^{N-1} (N - 2\hat{a})^2 \alpha_{\hat{a}} h_{2j}^{\hat{a}} \Bigg)
\end{aligned}
$$

$$
= -\frac{4}{N} \sum_{j} \sum_{\hat{a}=1}^{N-1} \alpha_{\hat{a}} \left( 1 - \frac{2\hat{a}}{N} - \frac{2}{N} \sum_{a=1}^{N-1} \alpha_{a,\hat{a}} h_{2j-1}^a \right) h_{2j}^{\hat{a}} \left( 1 - \frac{2\hat{a}}{N} - \frac{2}{N} \sum_{b=1}^{N-1} \alpha_{b,\hat{a}} h_{2j+1}^b \right) \tag{A14}
$$

Finally, note that taking (A12) and doing another commutator gives the Dolan-Grady relation:

$$
\Big[A_0, \big[A_0, [A_0, A_1]\big]\Big] = \left(\frac{4}{N}\right)^4 \left( 2\hat{a}(N - \hat{a}) + (N - 2\hat{a})^2 + 2\hat{a}(N - \hat{a}) \right)
$$

$$
\times \sum_{j=1}^{L} \sum_{a,\hat{a}=1}^{N-1} \alpha_{a,\hat{a}} \alpha_{\hat{a}} \left( h_{2j-1}^a h_{2j}^{\hat{a}} - h_{2j}^{\hat{a}} h_{2j+1}^a \right)
$$

$$
= 16 \big[A_0, A_1\big]. \tag{A15}
$$

A.3.1. *Proof of* (A11). Consider the following action:

$$
\sum_{a,b=0}^{N-1} \omega^{ka} \omega^{lb} X^{a+b} \left| v^{(j)} \right\rangle = \sum_{a,b=0}^{N-1} \omega^{ka+lb-ja-jb} \left| v^{(j)} \right\rangle = N^2 \delta_{k,j} \delta_{l,j} \left| v^{(j)} \right\rangle. \tag{A16}
$$

Using geometric series and the vanishing of the previous double sum for $k \neq l$ yields

$$
\sum_{a,b=1}^{N-1} \alpha_{a,\hat{a}} \alpha_{b,\hat{a}} X^{a+b} = \sum_{k,l=0}^{\hat{a}-1} \sum_{a,b=1}^{N-1} \omega^{ka+lb} X^{a+b} = \sum_{k,l=0}^{\hat{a}-1} \sum_{a,b=0}^{N-1} \omega^{ka+lb} X^{a+b} - 2\hat{a} \sum_{a=1}^{N-1} \sum_{k=0}^{\hat{a}-1} \omega^{ka} X^a - \hat{a}^2
$$

$$
= \hat{a}(N - \hat{a}) + (N - 2\hat{a}) \sum_{a=1}^{N-1} \sum_{k=0}^{\hat{a}-1} \omega^{ka} X^a. \tag{A17}
$$

A.4. **Connecting the two approaches.** In this section we show that the formula for $A_{-1}$ written in terms of $\widehat{\Phi}_n$ is the same as the expression (A14). In particular, we show:

$$
\left( \prod_{r=0}^{m-1} \widehat{\Phi}_j^{(r)} \right) = \widehat{S}_j^{(m)} \equiv 1 - \frac{2m}{N} - \frac{2}{N} \sum_{m'=1}^{N-1} \frac{1}{1 - \omega^{m'}} (1 - \omega^{mm'}) X_j^{m'}. \tag{A18}
$$

First, notice that the identity (A11) yields

$$
\left( S_j^{(m)} \right)^2 = \left( \widehat{S}_j^{(m)} \right)^2 = 1 \tag{A19}
$$

for any $m, j$. Thus the dressing terms are operators squaring to a constant. Indeed,

$$
\left( 1 - \frac{2}{N} \sum_{m'=0}^{N-1} \omega^{(r-a)m'} \right) \left| v_j^{(a)} \right\rangle = (1 - 2\delta_{r-a,0}) \left| v_j^{(a)} \right\rangle = (-1)^{\delta_{r-a,0}} \left| v_j^{(a)} \right\rangle = \widehat{\Phi}_j^{(r)} \left| v_j^{(a)} \right\rangle. \tag{A20}
$$

Using the geometric series $\sum_{r=0}^{m-1} \omega^{rm'} = \frac{1 - \omega^{mm'}}{1 - \omega^{m'}}$ yields

$$
\widehat{S}_j^{(m)} \left| v_j^{(a)} \right\rangle = \left( 1 - \sum_{r=0}^{m-1} \frac{2}{N} \sum_{m'=0}^{N-1} \omega^{rm'} X_n^{m'} \right) \left| v_j^{(a)} \right\rangle = \left( 1 - 2 \sum_{r=0}^{m-1} \delta_{r-a,0} \right) \left| v_j^{(a)} \right\rangle. \tag{A21}
$$

Comparing with (A20) and noting that $m < N$ yields (A18). One can also derive the right-hand-side of (A18) from the left-hand-side by taking products of (A20) for different values of $r$ and using (A16).

Analogous identities hold with the replacement $X_n \to Z_n^{-1} Z_{n+1}$; the latter gives a sign, but one that depends on the difference of the states on the two sites as in (A8). In particular,

$$\prod_{r=0}^{m-1} \Phi_{j-1,j}^{(r)} = S_{j-1,j}^{(m)} \equiv \left( 1 - \frac{2m}{N} - \frac{2}{N} \sum_{m'=1}^{N-1} \alpha_{m'} \left( 1 - \omega^{mm'} \right) Z_{j-1}^{-m'} Z_j^{m'} \right), \tag{A22}$$

leading to (17).

### Appendix B. String order and parity transformations

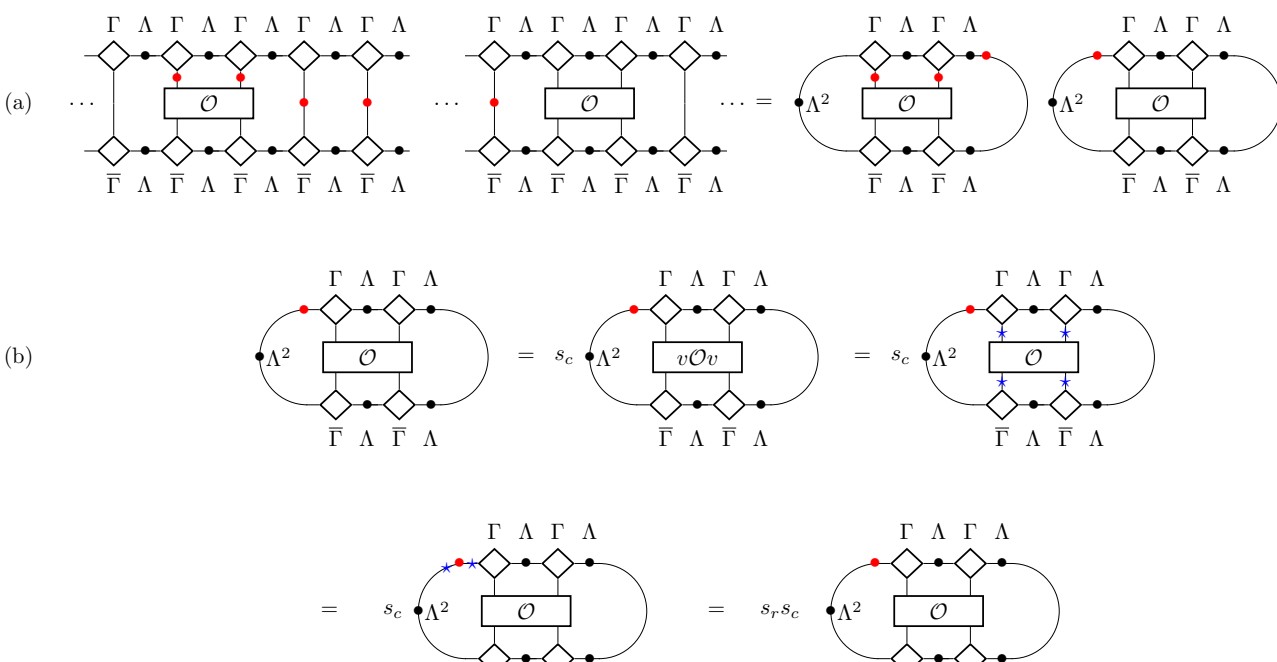

FIGURE 8. Graphical identities for an MPS tensor with $\mathbb{Z}_2 \times \mathbb{Z}_2$ symmetry generated by $\prod_k u_k$ and $\prod_k v_k$: (a) the transfer matrix in canonical form (b) symmetry fractionalisation of $u$ (c) symmetry fractionalisation of $v$. Recall that on the bonds $[U, \Lambda] = [V, \Lambda] = 0$.

FIGURE 9. Graphical proof, following [65], that long-range string order with a charged endpoint is non-vanishing only in the SPT phase for $\mathbb{Z}_2 \times \mathbb{Z}_2$ symmetry. In the text we generalise this to the $\mathbb{Z}_2 \times \mathbb{Z}_2^{\mathrm{CPT}}$ case. The $\bullet$ indicate the symmetry $u$, and the $\star$ indicate the symmetry $v$ (in the appropriate representation for either physical or bond indices). (a) The string order written as a tensor contraction using the MPS ground state, we apply Fig. 8(a) to simplify this to two local tensor contractions (the equality holds up to exponentially small corrections in the string length). Using Fig. 8(b) these terms are identical. (b) Analysing the local tensor contraction using the charge of the end-point $v\mathcal{O}v = s_c \mathcal{O}$ and the SPT phase $VUV = s_r U$. If the signs do not match then the string-order must vanish.

In this appendix we will show, following Ref. [65], that the charge of endpoints of string operators with long-range order reveals the SPT order for a $\mathbb{Z}_2 \times \mathbb{Z}_2^{\mathrm{CPT}}$ symmetry. The usual arguments are modified due to the $\mathbb{Z}_2^{\mathrm{CPT}}$ acting non-trivially on the lattice (through the parity transformation). In Figs. 8 and 9 we review the usual argument for the $\mathbb{Z}_2 \times \mathbb{Z}_2$ case graphically. Our analysis follows

similar steps, but we use tensor notation to make the index transposition explicit. In this appendix we allow a more general representation of $\mathbb{Z}_2 \times \mathbb{Z}_2^{\mathrm{CPT}}$ than needed for the models in the main text. For technical reasons, we also assume translation invariance.

First, let us define a *symmetry flux* for the $\mathbb{Z}_2$ symmetry $\prod_j u_j$ by

$$\Sigma_n = \prod_{j \leq n} u_j \mathcal{O}_{n+1,\ldots,n+k} \ , \tag{B1}$$

where we require that the end-point $\mathcal{O}$ is hermitian, and note that $u_n^2 = 1 \implies u_n^\dagger = u_n$. The end-point is required to be hermitian so that there is no remaining phase freedom that would leave the charge under the anti-unitary CPT symmetry ill-defined [6]. The symmetry flux for a particular ground state is the half-infinite symmetry string with end-point such that the two-point function has the slowest possible decay—for gapped phases of matter this is the case with long-range order.

For an end-point operator $\mathcal{O}$ supported on $k$ sites, we require that $\mathcal{O}u^{\otimes k} = u^{\otimes k}\mathcal{O}$ (this is the case in the main text, with $k = 2$). Note that this commutativity is equivalent to $\Sigma_n$ being neutral under the $\mathbb{Z}_2$ symmetry. Moreover, this is equivalent to $\mathcal{O}u^{\otimes k}$ being hermitian, implying that the string-order $\Sigma_1 \Sigma_n$ is hermitian.

We are interested in end-point operators with a charge under the other symmetry $\mathbb{Z}_2^{\mathrm{CPT}}$, where the charge is relative to multiplication by $u^{\otimes k}$. That is,

$$\mathcal{CPK} \, \mathcal{O} \, \mathcal{CPK} = s_{\mathrm{c}} u^{\otimes k} \mathcal{O} \equiv s_{\mathrm{c}} \tilde{\mathcal{O}} \qquad s_{\mathrm{c}} = \pm 1 \ , \tag{B2}$$

where we define $\tilde{\mathcal{O}} = u^{\otimes k}\mathcal{O}$ for convenience. As in the main text, we note that the parity symmetry $\mathcal{P}$ will translate the support of the operator in general. We suppose $\mathcal{C} = \prod_j C_j$ is a product of an on-site unitary involution that is real in the $Z$-basis[3]. This operation corresponds to applying the symmetry and then multiplying the inverted (B1) by the $\mathbb{Z}_2$ symmetry $\prod_j u_j$ so that we can compare the end-point of the same half-infinite string. This multiplication by $u^{\otimes k}$ also arises naturally in the MPS formalism, as we demonstrate in the proof below.

The string order is the two-point correlator of the (hermitian) symmetry flux (B1)

$$\langle \Sigma_1 \Sigma_M \rangle = \langle \tilde{\mathcal{O}}_{1,2,\ldots,k} \left( \prod_{j=k+1}^{M} u_j \right) \mathcal{O}_{M+1,\ldots,M+k} \rangle. \tag{B3}$$

Let $s_{\mathrm{c}}$ be the charge of the end-point and let $s_{\mathrm{r}}$ be the invariant charge of the projective representation of $\mathbb{Z}_2 \times \mathbb{Z}_2^{\mathrm{CPT}}$ on the virtual degrees of freedom ($UV = s_{\mathrm{r}}VU$). Assuming a symmetric ground state, we have the selection rule:

$$\lim_{M,L \to \infty} \langle \Sigma_1 \Sigma_M \rangle = x^2 \qquad \text{for } x \in \mathbb{R}$$

$$\text{and} \qquad x = s_{\mathrm{c}} s_{\mathrm{r}} x. \tag{B4}$$

This means that the long-range string order is non-vanishing only if $s_{\mathrm{c}} = s_{\mathrm{r}}$, giving us a method of detecting the SPT phase with a lattice observable.

B.1. **Proof of string order selection rule.** We derive (B4) using the MPS formalism; using the area law for ground states of gapped local Hamiltonians in 1D to justify using these results more generally [61, 105]. The outline of the proof is roughly given in Fig. 9; the complication is the inclusion of the time-reversal and parity transformations. Together, these act on the MPS tensors to take $\Gamma \to \Gamma^\dagger$, allowing us to use the symmetry fractionalisation of $\mathcal{C}$ as in Fig. 9(b). Our argument uses translation-invariance, and we will use this translation symmetry when we act with the $\mathcal{P}$ symmetry on the local tensor contraction (denoted $x$ below)—in particular, we will choose $\mathcal{P}$ to invert about the central bond (or site) of the support of $\tilde{\mathcal{O}}$.

---

[3]Without loss of generality we can take our inversion $\mathcal{P}$ to be real. We expect a similar argument to go through if we consider the more general case $\mathcal{CPK} \, \mathcal{O} \, \overline{\mathcal{CPK}} = \pm \tilde{\mathcal{O}}$, since the $\mathbb{Z}_2$ implies $(\mathcal{CP})^\dagger = \overline{\mathcal{CP}}$. Moreover, an analogous charge and selection rule applies in the case where we have a $\mathbb{Z}_2^P$ or $\mathbb{Z}_2^{CP}$ symmetry that includes a parity transformation but does not include time-reversal.

Take a (translation-invariant) MPS representation of the ground state, with tensors $\mathcal{A}_j$ in canonical form. We can write the correlator in terms of the generalised transfer matrix $E_{\mathcal{X}} = \sum_{j,j'=0}^{N-1} \mathcal{X}_{j'j} \mathcal{A}_j \otimes \mathcal{A}_{j'}^\dagger$ (with the natural generalisation to operators supported on multiple sites). We then have

$$\langle \Sigma_1 \Sigma_M \rangle = \text{tr} \left( E_{\mathbb{I}}^{L-M-k} E_{\tilde{\mathcal{O}}} E_u^{M-k} E_{\mathcal{O}} \right), \tag{B5}$$

which can be simplified (up to exponentially small corrections) for large chain length, $L$, and large string length, $M$, as

$$\langle \Sigma_1 \Sigma_M \rangle \simeq \underbrace{\left\langle \Lambda^2 \middle| E_{\tilde{\mathcal{O}}} \middle| U \right\rangle}_{x} \underbrace{\left\langle \Lambda^2 U \middle| E_{\mathcal{O}} \middle| \mathbb{I} \right\rangle}_{y}. \tag{B6}$$

This follows from the implicit assumption that the unique dominant eigenvalue of $E_{\mathbb{I}}$ is equal to one, and that $\sum_{j'} u_{jj'} \mathcal{A}_{j'}$ fractionalises to $U \mathcal{A}_j U$ on the bonds (we can fix the phase so that $U = U^\dagger$ since we have a representation of $\mathbb{Z}_2$). This can be seen graphically for $k = 2$ in Fig. 9(a).

The non-vanishing of the string order means that neither of the two factors in (B6) vanishes. We will show first that these factors are both equal to the same real number. That is, $x = y \in \mathbb{R}$ (the equality is straightforward to see graphically, as in Fig. 9(a)). We then show that a charged endpoint will cause $x$ to vanish unless there is a non-trivial projective representation of $\mathbb{Z}_2 \times \mathbb{Z}_2^{\text{CPT}}$ on the bonds. This is in line with the string order for an on-site $\mathbb{Z}_2 \times \mathbb{Z}_2$ symmetry [65].

Define the matrix $\tilde{M}^{\gamma\delta} = \left[ \left\langle \Lambda^2 \middle| E_{\tilde{\mathcal{O}}} \right. \right]^{\gamma\delta}$ then

$$\tilde{M}^{\gamma\delta} = \sum \Lambda_\alpha^2 \mathcal{A}_{j_1}^{\alpha\beta_1} \overline{\mathcal{A}}_{j_1'}^{\alpha\beta_1'} \left( \prod_{m=1}^{k-2} \mathcal{A}_{j_{m+1}}^{\beta_m \beta_{m+1}} \overline{\mathcal{A}}_{j_{m+1}'}^{\beta_m' \beta_{m+1}'} \right) \mathcal{A}_{j_k}^{\beta_k \gamma} \overline{\mathcal{A}}_{j_k'}^{\beta_k \delta} \, \tilde{\mathcal{O}}_{j_1 \ldots j_k, j_1' \ldots j_k'} \tag{B7}$$

(where we sum over all indices except $\gamma$ and $\delta$). Then $x = \left\langle \Lambda^2 \middle| E_{\tilde{\mathcal{O}}} \middle| U \right\rangle = \text{tr}(\tilde{M} U)$. Now, we have that $\overline{x} = \overline{\langle \Lambda^2 | E_{\tilde{\mathcal{O}}} | U \rangle} = \text{tr}(\tilde{M}^\dagger U)$. Using that $\tilde{\mathcal{O}} = \tilde{\mathcal{O}}^\dagger$ we have that that $\tilde{M}^\dagger = \tilde{M}$ and so $x \in \mathbb{R}$. We also have from symmetry fractionalisation that $\sum_\beta \mathcal{A}_j^{\alpha\beta} U^{\beta\gamma} = \sum_{\beta,\tilde{j}} u_{j,\tilde{j}} U^{\alpha\beta} \mathcal{A}_{\tilde{j}}^{\beta\gamma}$. We can then move the $U$ to the left, at each step applying $u$ to the physical index. This gives us that $x = \text{tr}(\tilde{M} U) = \text{tr}(M)$ where $M^{\gamma\delta} = \left[ \left\langle \Lambda^2 U \middle| E_{\mathcal{O}} \right. \right]^{\gamma\delta}$; i.e.

$$M^{\gamma\delta} = \sum \Lambda_\alpha^2 U^{\alpha\alpha'} \mathcal{A}_{j_1}^{\alpha'\beta_1} \overline{\mathcal{A}}_{j_1'}^{\alpha\beta_1'} \left( \prod_{m=1}^{k-2} \mathcal{A}_{j_{m+1}}^{\beta_m \beta_{m+1}} \overline{\mathcal{A}}_{j_{m+1}'}^{\beta_m' \beta_{m+1}'} \right) \mathcal{A}_{j_k}^{\beta_k \gamma} \overline{\mathcal{A}}_{j_k'}^{\beta_k \delta} \, \mathcal{O}_{j_1 \ldots j_k, j_1' \ldots j_k'}. \tag{B8}$$

We recognise the trace of $M$ as $y$, and hence both factors in (B6) are equal to $x \in \mathbb{R}$.

The symmetry fractionalisation of $\mathbb{Z}_2^{\text{CPT}}$ is

$$\sum_{j'=0}^{N-1} C_{j,j'} \, (\Gamma_{j'}^{\alpha,\beta})^\dagger = V \, \Gamma_j^{\alpha,\beta} \, V, \tag{B9}$$

where we decompose $\mathcal{A}_j^{\alpha,\beta} = \Gamma_j^{\alpha,\beta} \Lambda_\beta$. Again, since $C^2 = \mathbb{I}$ we can fix the phase so that $V = V^\dagger$. Just as in the on-site $\mathbb{Z}_2 \times \mathbb{Z}_2$ case, $UV = s_{\text{r}} VU$ is a gauge invariant phase that determines the SPT order.

Now, using $x = \overline{x} = \text{tr}(M)$ and $U = U^\dagger$ we have

$$x = \sum \Lambda_\alpha^2 U^{\alpha'\alpha} \overline{\mathcal{A}}_{j_1}^{\alpha'\beta_1} \mathcal{A}_{j_1'}^{\alpha\beta_1'} \left( \prod_{m=1}^{k-2} \overline{\mathcal{A}}_{j_{m+1}}^{\beta_m \beta_{m+1}} \mathcal{A}_{j_{m+1}'}^{\beta_m' \beta_{m+1}'} \right) \overline{\mathcal{A}}_{j_k}^{\beta_k \gamma} \mathcal{A}_{j_k'}^{\beta_k \gamma} \, \overline{\mathcal{O}}_{j_1 \ldots j_k, j_1' \ldots j_k'}. \tag{B10}$$

Inserting $\mathbb{I} = \mathcal{P}^2$ on the physical indices we have:

$$x = \sum \mathcal{A}^{\dagger \alpha\beta_1}_{j_1} \mathcal{A}^{T \alpha\beta_1'}_{j_1'} \left( \prod_{m=1}^{k-2} \mathcal{A}^{\dagger \beta_m \beta_{m+1}}_{j_{m+1}} \mathcal{A}^{T \beta_m' \beta_{m+1}'}_{j_{m+1}'} \right) \mathcal{A}^{\dagger \beta_k \gamma}_{j_k} \mathcal{A}^{T \beta_k \gamma'}_{j_k'} U^{\gamma\gamma'} \Lambda_\gamma^2 \, (\mathcal{P} \overline{\mathcal{O}} \mathcal{P})_{j_1 \ldots j_k, j_1' \ldots j_k'}. \tag{B11}$$

Finally inserting $\mathbb{I} = \mathcal{C}^2$ gives

$$x = \sum \Lambda_\alpha^2 \mathcal{A}_{j_1}^{\alpha\beta_1} \overline{\mathcal{A}}_{j_1'}^{\alpha\beta_1'} \left( \prod_{m=1}^{k-2} \mathcal{A}_{j_{m+1}}^{\beta_m \beta_{m+1}} \overline{\mathcal{A}}_{j_{m+1}'}^{\beta_m' \beta_{m+1}'} \right) \mathcal{A}_{j_k}^{\beta_k \gamma} \overline{\mathcal{A}}_{j_k'}^{\beta_k \gamma'} (VUV)^{\gamma\gamma'} (\mathcal{C}\mathcal{P} \overline{\mathcal{O}} \mathcal{P}\mathcal{C})_{j_1 \ldots j_k, j_1' \ldots j_k'}. \tag{B12}$$

Comparing this expression to $x = \operatorname{tr}(\tilde{M}U)$ we see that $x = s_c s_r x$ as claimed.

B.2. **String order for $N = 4$.** In Section 5.3 we write the string order (47) in terms of the simple string correlators:

$$S_{a,b} = \lim_{M,L\to\infty} \langle Z_1^a Z_2^{-a} X_2^2 \ldots X_M^2 Z_M^b Z_{M+1}^{-b} \rangle. \tag{B13}$$

Multiplying out the terms that appear for $N = 4$, the string order (47) is equal to

$$\lim_{M,L\to\infty} \langle \tilde{\mathcal{O}}_{0,1} \left( \prod_{j=2}^{M-1} X_j^{N/2} \right) \mathcal{O}_{M,M+1} \rangle = \frac{i}{2} \left( S_{-1,-1} - S_{1,1} \right) + \frac{1}{2} \left( S_{-1,1} + S_{1,-1} \right). \tag{B14}$$

Using the same MPS transfer matrix arguments as in the previous subsection we have that $S_{a,b}$ is equal to $M^{(a)}\tilde{M}^{(b)}$, where

$$M^{(a)} = \sum \Lambda_\alpha^2 \mathcal{A}_{j_1}^{\alpha\beta_1} \overline{\mathcal{A}}_{j_1'}^{\alpha\beta_1'} \mathcal{A}_{j_2}^{\beta_1\gamma} \overline{\mathcal{A}}_{j_2'}^{\beta_1'\delta} \; Z_{j_1,j_1'}^a (Z^{-a}X^2)_{j_2,j_2'} U^{\gamma\delta} \tag{B15}$$

$$\tilde{M}^{(b)} = \sum \Lambda_\alpha^2 U^{\alpha\alpha'} \mathcal{A}_{j_1}^{\alpha\beta_1} \overline{\mathcal{A}}_{j_1'}^{\alpha'\beta_1'} \mathcal{A}_{j_2}^{\beta_1\gamma} \overline{\mathcal{A}}_{j_2'}^{\beta_1'\delta} \; (X^2 Z^b)_{j_1,j_1'} Z_{j_2,j_2'}^{-b}. \tag{B16}$$

$U$ is the fractionalised $\mathbb{Z}_2$ symmetry that acts as $X^2$ on physical indices. Using symmetry fractionalisation and that $X^2 Z^{\pm 1} = -Z^{\pm 1} X^2$ we have that $M^{(a)} = -\tilde{M}^{(a)}$.

Conjugating $M^{(a)}$ amounts to taking the hermitian conjugate of the physical operator. Since we have $(Z_1^a Z_2^{-a} X_2^2)^\dagger = -Z_1^{-a} Z_2^a X_2^2$, we conclude that $\overline{M}^{(a)} = -M^{(-a)}$. Putting this together

$$\lim_{R\to\infty} \langle \tilde{\mathcal{O}}_{0,1} \left( \prod_{j=2}^{R-1} X_j^{N/2} \right) \mathcal{O}_{R,R+1} \rangle = -\operatorname{Im}(M^{(-1)}\overline{M}^{(1)}) + \frac{1}{2} \left( |M^{(-1)}|^2 + |M^{(1)}|^2 \right)$$

$$= \frac{i}{2} \left( S_{-1,-1} - S_{1,1} \right) + S_{-1,1}. \tag{B17}$$

Using the fixed-point MPS (34), one can show explicitly that $\operatorname{Re}(M^{(1)}) = -\operatorname{Im}(M^{(1)})$. At the fixed-point this means that the string correlator $1 = \lim_{R\to\infty} \langle \tilde{\mathcal{O}}_{0,1} \left( \prod_{j=2}^{R-1} X_j^{N/2} \right) \mathcal{O}_{R,R+1} \rangle = 2S_{-1,1}$. According to our numerics, this relationship continues to hold away from the fixed point. It would be interesting to establish this analytically using the projective representations.

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
