# Peer review of "Pivoting through the chiral-clock family"

_SciPost Physics_

## Round 1 · Referee Report · Anonymous (Referee 1) · 2024-9-10

Strengths

1-Detailed study of the phase diagrams for the Hamiltonian $H(\alpha,\beta,\gamma)$ based on a variety of numerical probes

Weaknesses

2-The notions of SPT and RSPT could be introduced more clearly

Report

This is a very interesting and complete study of the physics of a family of $N$ states chiral clock models based on the Onsager algebra. A first observation is that the latter can be used as a pivot between the different Onsager generators, enabling one to construct the ground state of generators $A_l$ from the known ones in the cases $l=0,1$. Since the pivot procedure acts as an SPT entangler, some of the constructed Hamiltonians have SPT order (or a weaker form thereof known as RSPT).
The authors then move on to a detailed numerical investigation of the phase diagram of Hamiltonians constructed from $A_0,A_1,A_2$, generalising a known result for $N=2$. The phase diagrams display a rich variety of transitions between gap, gapless and (R)SPT phases.

Despite some blind spots in the phase diagram (some of which are the subject of a forthcoming companion paper), I think this work decisively meets the criteria for publication in SciPost. I however have some comments, listed below. In particular, it feels that the notions of SPT and RSPT could be better introduced for someone not familiar with the subject. Since the case $N=2$ is well-known and an excellent toy-model for the SPT physics, I would suggest to start Section 4 with a brief review of the $N=2$ physics. In particular, it would be useful to see the ground state of $A_2$ written explicitly, in order to present a first concrete encounter with symmetry fractionalisation.

Requested changes

1- Section 2: all the pivots considered here map between $A_m$ generators. Is there a reason why the authors did not consider pivots between the generators $G_m$ ?

2- As explained above, I suggest to start Section 4 with a brief review of the known $N=2$ physics, introducing in particular the spin-1/2 cluster model, SPT and symmetry fractionalisation

3- Looking at the phase diagram for $N=2,3,4$ (Fig. 3), one is lead to wonder whether the gapless region keeps on expanding for larger $N$. Could the authors comment ? Is there any insight about the $N\to \infty$ limit?

Minor requests :

4- above eq. (23), in the definition of $\mathcal{V}$ : is $\mathcal{K}$ inside or outside the product ? Please add parentheses to resolve the ambiguity.

5- there is a typo after eq. (29) : "The $N$ ground states of $E_1$" : $E_1$ should be replaced by $A_1$.

Recommendation

Publish (easily meets expectations and criteria for this Journal; among top 50%)

  • validity: top
  • significance: high
  • originality: high
  • clarity: good
  • formatting: excellent
  • grammar: perfect

Author:  Nick Jones  on 2025-01-11  [id 5106]

(in reply to Report 1 on 2024-09-10)

Thank you for the careful reading of our manuscript, positive feedback, and the suggestions where we can improve the presentation. We have addressed the definition of SPT more carefully in the introduction, as detailed in the response to Referee 2. We have also added the following text to the introduction "For our purposes, we say that a ground state is an RSPT when the dominant Schmidt eigenvalues have a degeneracy due to a non-trivial linear irreducible representation of the non-abelian symmetry." where we first discuss RSPTs. This should clarify the notion at an early stage before the more detailed discussion in Section 4.

1- Section 2: all the pivots considered here map between Am generators. Is there a reason why the authors did not consider pivots between the generators Gm ?

We chose to study the Am family as in the N=2 case this is a well understood family with rich phase diagram. The Gm themselves would have trivial pivot relations since they all commute, but the Gm would behave nicely under pivots generated by An. This would be an interesting subject for future study.

2- As explained above, I suggest to start Section 4 with a brief review of the known N=2 physics, introducing in particular the spin-1/2 cluster model, SPT and symmetry fractionalisation

Thank you for this suggestion. We have included a new Section 4.1 that does exactly this, and we choose to present the fractionalisation of Z_2 x Z_2^T so that this is not simply a special case of our general analysis later.

3- Looking at the phase diagram for N=2,3,4 (Fig. 3), one is lead to wonder whether the gapless region keeps on expanding for larger N. Could the authors comment ? Is there any insight about the N→∞ limit?

This is a nice observation, but we are not aware of any results for N>4 that would clarify whether this trend continues, nor results for N->\infty. It would be very interesting to understand this, and likely the first place to look would be at the first transition point on the line A_0 + \lambda A_1 as a function of N.

4- above eq. (23), in the definition of V : is K inside or outside the product ? Please add parentheses to resolve the ambiguity.

We have added these.

5- there is a typo after eq. (29) : "The N ground states of E1" : E1 should be replaced by A1.

Thank you for pointing this out - we have made this replacement.

---

## Round 1 · Referee Report · Anonymous (Referee 2) · 2024-11-27

Strengths

1- interesting set of models 2- good calculations 3- timely topic

Weaknesses

1- the presentation is not appropriate, as the paper is too hard to read and not precise enough about the crucial concept it discusses.

Report

In this paper, the authors consider a family of integrable Hamiltonians that are formed out of linear combinations of (a representation of) Onsager algebra generators. They show that these are related to each other by a “pivot” procedure, and that this therefore helps finding Hamiltonians in the “SPT” phase.

The paper is interesting and seems to produce good results and nice and fundamental ideas. The idea of using the Onsager algebra in order to study the Hamiltonians and especially their SPT property is very interesting. I believe this paper can be published in Scipost Physics in some form.

However in the current form I find the paper very difficult to read. It lacks the basic definitions of some of the main concepts discussed (like “SPT phase” itself); the general discussion in the introduction and section 2 are only marginally useful, and the reader must try to guess what is happening as the more precise results for the specific models are then discussed. I don’t have specific comments on the physics or mathematics, which I believe is correct and deep. But I believe the presentation should be very greatly improved.

The authors should provide much better explanations about what are “pivot relation” and “SPT phase”, and how a Hamiltonian can be “in a phase” (instead of a state being in a phase, as is usual in statistical mechanics). See the comments below for more (related and non-related) questions and requirement for improved explanations.

Introduction: What does “any Hamiltonians giving rise to this algebra” mean? What does it mean that a Hamiltonian “gives rise” to an algebra? Perhaps that the Hamiltonian lies within this algebra (a specific representation of it)? Also, it seems that the statement that any two Hamiltonians satisfy “the same pivot relations” is not so clear. I thought the pivot relation was to produce SPT phases, but not all cases do produce SPT phases (from the abstract, and from the rest of the paper). What is, then, really, a “pivot relation”? This is also not clear at this stage.

Page 3: the statement that (2) results in an infinite-dimensional algebra as stated (with two discrete families) is in fact not very meaningful. Its only meaning is that the Dolan-Grady relations are not enough to make the algebra finite-dimensional. Indeed any two generators, without additional relations, lead to an algebra that is, at most, countably-dimensional, which can be organised into any number of discrete families as one wishes. Could the author make the statement more precise? Perhaps define the $A_l$ and $G_m$, or give their main properties? Or just refer to (4) already?

Page 4, beginning of 2.2: what is the meaning of “trivial ground state”? Please give some explanations there.

Page 4, after eq (6): could the author explain in some words with basic equations what SPT order means for $H_{SPT}$? What is the “SPT order”? What is “an SPT Hamiltonian”? How is the ground state of $H_{SPT}$ non-trivial? What is an “SPT” entangler”? What is the meaning of a Hamiltonian “being a non-trivial SPT phase”? This is strange for somebody coming from standard statistical mechanics, as normally the terminology “phase” is reserved for certain thermodynamic states of a given Hamiltonian, and not for characterising a Hamiltonian. More explanations about these fundamental concepts would make the paper more readable.

Page 4, 2.3: what is the meaning of “behave nicely” here? Please be more precise.

Page 5, after eq 12: why are the two families related by KW duality? Have the authors prove this? More explanations would be good.

Page 5, paragraph around eq 13: it is not clear to me how phase structures are obtained from these relations. The explanation is not so convincing: that all $A_{2k}$ are non-trivial SPT phases seem a strong requirement, and the conclusion that $A_k/4$ is a pivot seems then like a tautology.

Page 9, beginning of sect 4: “we showed in section 3.4 that…” What “such a form” does the ground stat of $A_2$ have? The unitary transformation of a “trivial” hamiltonian (sum of commuting terms, etc)? Please be more specific - I don’t think SPT phase was shown already.

Page 9, bottom of the page: it is not clear (at that point) what is meant by a representation of symmetry generators “in the ground state”. The ground state is a single state (or do the authors mean degeneracies? The ground space?), and is usually not enough to form by itself a representation. Do the authors mean the representation on the MPS form of the ground state, within the equivalence class of MPS matrices for that state?

Requested changes

See the report

Recommendation

Ask for major revision

  • validity: top
  • significance: high
  • originality: good
  • clarity: low
  • formatting: perfect
  • grammar: excellent

Author:  Nick Jones  on 2025-01-11  [id 5107]

(in reply to Report 2 on 2024-11-27)

In this paper, the authors consider a family of integrable Hamiltonians that are formed out of linear combinations of (a representation of) Onsager algebra generators. They show that these are related to each other by a “pivot” procedure, and that this therefore helps finding Hamiltonians in the “SPT” phase.

The paper is interesting and seems to produce good results and nice and fundamental ideas. The idea of using the Onsager algebra in order to study the Hamiltonians and especially their SPT property is very interesting. I believe this paper can be published in Scipost Physics in some form.

However in the current form I find the paper very difficult to read. It lacks the basic definitions of some of the main concepts discussed (like “SPT phase” itself); the general discussion in the introduction and section 2 are only marginally useful, and the reader must try to guess what is happening as the more precise results for the specific models are then discussed. I don’t have specific comments on the physics or mathematics, which I believe is correct and deep. But I believe the presentation should be very greatly improved.

We are grateful for this considered reading of our paper, positive response to the key results, and for the detailed comments regarding improved presentation. We agree that the key notions of SPT physics are very important to a good understanding of the paper, and we have added several clarifications to the text to aid the reader. As well as these clarifications, we have added a new section 4.1 that introduces the concept of symmetry fractionalisation in a simple example. We address the comments in turn below, and hope that with these changes our manuscript is now suitable for publication in SciPost Physics.

The authors should provide much better explanations about what are “pivot relation” and “SPT phase”, and how a Hamiltonian can be “in a phase” (instead of a state being in a phase, as is usual in statistical mechanics). See the comments below for more (related and non-related) questions and requirement for improved explanations.

We address the points around SPT phases in the new footnote 1 which reads "A Hamiltonian belonging to the trivial phase has a unique ground state that can be smoothly connected to a product state without breaking symmetry. Hamiltonians belonging to non-trivial SPT phases have unique ground states in the absence of boundaries but cannot be adiabatically connected to a product state along a symmetric path. Note that we consider the zero temperature phase diagram and so classifying gapped `parent' Hamiltonians and their ground states is equivalent [10]. Key notions are reviewed in [2, 5, 11–13]."

We have added a comment defining RSPT in the introduction.

The pivot relation is addressed in the next point.

We also further clarify SPT order corresponding to non-trivial projective representations at the beginning of Section 4.

Introduction: What does “any Hamiltonians giving rise to this algebra” mean? What does it mean that a Hamiltonian “gives rise” to an algebra? Perhaps that the Hamiltonian lies within this algebra (a specific representation of it)? Also, it seems that the statement that any two Hamiltonians satisfy “the same pivot relations” is not so clear. I thought the pivot relation was to produce SPT phases, but not all cases do produce SPT phases (from the abstract, and from the rest of the paper). What is, then, really, a “pivot relation”? This is also not clear at this stage.

We have amended this to " Any pair of Hamiltonians generating an Onsager algebra will satisfy the same pivot relations” clarifying our meaning.

The corresponding paragraph "The Onsager algebra.... symmetry group “ has been substantially edited to clarify how we view pivot relations in the context of this paper. We make this particular clarification as we wish to also consider the RSPT case. We clarify this again in Section 2.2.

Page 3: the statement that (2) results in an infinite-dimensional algebra as stated (with two discrete families) is in fact not very meaningful. Its only meaning is that the Dolan-Grady relations are not enough to make the algebra finite-dimensional. Indeed any two generators, without additional relations, lead to an algebra that is, at most, countably-dimensional, which can be organised into any number of discrete families as one wishes. Could the author make the statement more precise? Perhaps define the Al and Gm, or give their main properties? Or just refer to (4) already?

We have amended the text to read "imposing these identities, now known as the Dolan-Grady conditions, results in an infinite-dimensional Lie algebra, canonically represented by a set of generators..." We hope this makes clear that this is just a standard choice; the precise definitions of these generators follows shortly after.

Page 4, beginning of 2.2: what is the meaning of “trivial ground state”? Please give some explanations there. Page 4, after eq (6): could the author explain in some words with basic equations what SPT order means for HSPT? What is the “SPT order”? What is “an SPT Hamiltonian”? How is the ground state of HSPT non-trivial? What is an “SPT” entangler”? What is the meaning of a Hamiltonian “being a non-trivial SPT phase”? This is strange for somebody coming from standard statistical mechanics, as normally the terminology “phase” is reserved for certain thermodynamic states of a given Hamiltonian, and not for characterising a Hamiltonian. More explanations about these fundamental concepts would make the paper more readable.

We have amended “trivial ground state” to "trivial (product state) ground state" .

We have addressed the general points about SPT phases directly in the introduction, see the new footnote 1. The SPT entangler is introduced in the first paragraph.

We have considered adding further definitions, but feel that too many notes of this kind will not improve readability as it will break up the flow of our arguments. We expect many readers will have familiarity with these notions and have provided several references for those wishing for more background.

Page 4, 2.3: what is the meaning of “behave nicely” here? Please be more precise.

We have amended "we can generate a family of Hamiltonians that behave nicely under pivoting" to "we show how pivoting generates a family of Hamiltonians with a simple closed form in terms of the Onsager generators."

Page 5, after eq 12: why are the two families related by KW duality? Have the authors prove this? More explanations would be good.

We agree that in the abstract setting after eq 12 this is not necessarily a KW duality, thank you for pointing this out. We have clarified this and added a reference for the KW duality in the chiral clock family.

Page 5, paragraph around eq 13: it is not clear to me how phase structures are obtained from these relations. The explanation is not so convincing: that all A2k are non-trivial SPT phases seem a strong requirement, and the conclusion that Ak/4 is a pivot seems then like a tautology.

We do not claim that all A2k are non-trivial SPT phases. We are pointing out that, for example, no A2k can ever have spontaneous symmetry breaking (unlike, say, A1). We agree that Ak/4 being a pivot when A2k is an SPT for some k is straightforward, however, we do not believe it is a tautology since the A2k are not themselves defined using these unitary rotations. We have edited this paragraph to make these points clearer, thank you for flagging this issue.

Page 9, beginning of sect 4: “we showed in section 3.4 that…” What “such a form” does the ground stat of A2 have? The unitary transformation of a “trivial” hamiltonian (sum of commuting terms, etc)? Please be more specific - I don’t think SPT phase was shown already.

We have amended this line to "In Section 3.4 we showed how the ground state of A2 indeed takes the form of a finite-depth local unitary transformation applied to a trivial state. "

We have now defined a trivial Hamiltonian above in e.g. footnote 1.

Page 9, bottom of the page: it is not clear (at that point) what is meant by a representation of symmetry generators “in the ground state”. The ground state is a single state (or do the authors mean degeneracies? The ground space?), and is usually not enough to form by itself a representation. Do the authors mean the representation on the MPS form of the ground state, within the equivalence class of MPS matrices for that state?

We have added the text " Concretely, this corresponds to the classification of non-trivial projective representations of the symmetry group on the bond indices of an MPS representation of the ground state." to the second paragraph of Section 4. This defines what we mean by these representations, and we think that the new Section 4.1 will further address this, by introducing the concept of symmetry fractionalisation in a simple example.

---

## Editorial Decision

resubmitted